# CaliBench: Are the Stochastic Dynamics of Video World Models Physically Calibrated?

## Abstract

Video world models are designed to approximate the stochastic *distribution* of physical outcomes through generative sampling, but existing benchmarks evaluate properties of each individual generation or aim to compare output distributions in a coarse grained way over an entire dataset, and do not investigate the fine grained aleatoric uncertainty of specific physical phenomena. We introduce **CaliBench**, a benchmark that tests this directly by scoring outcomes in a physically interpretable discrete space — a bin index, a die face, a suit, a colour — rather than in a learned feature space (e.g. that of FID), where we are able to directly calculate the distributional distance of the model outputs from a known reference distribution. This is possible because we carefully curate outcome spaces where the reference distribution is known in closed form (binomial Galton boards, Bernoulli forks, uniform dice/cards/lottery, a known-skewed European-roulette colour), which enables an exact calibration test which is not dependent on an empirical proxy. We decompose performance into two orthogonal axes that a single accuracy metric would conflate: *scorability* (the fraction of video generations producing a scoreable outcome) and *calibration* (the total variation distance from the reference distribution on the scoreable sample). We also test the statistical significance of miscalibration with a $\chi^2$ test; because calibration is its null hypothesis, the test can only evidence miscalibration, never calibration, and at $N=32$ generations per cell it reliably detects only large miscalibrations. We apply the protocol to nine scenes and six state-of-the-art image-to-video models (WAN-2.7, SeeDance-2.0, HappyHorse-1.0, Veo 3.1, Runway Gen-4.5, Cosmos3-Super) on 32 video generations each. Results reveal a consistent pattern: models concentrate output probability mass on a small subset of outcomes rather than reproducing the reference distribution. Most scene–model combinations are significantly miscalibrated, in the most extreme case collapsing entirely to a single outcome, for example Veo 3.1 on dice. On roulette, generated videos often leave the ball ambiguously placed, so several models also have low scorability. Performance varies sharply by scene rather than by model: no single model dominates across all nine scenes. We release the protocol and a metric (mean normalised total variation, mnTV) to enable comparison of new models with the results in this paper.

## 1 Introduction

Simulation theory fundamentally questions whether a computed environment can be statistically distinguished from reality (Bostrom, 2003). In generative modelling, a parallel problem emerges: a simulation often betrays itself not through catastrophic visual failures, but through subtle statistical anomalies—glitches or regularities that deviate from genuine randomness. If stochastic macroscopic events generated by a model, such as coin flips or roulette spins, follow distributions that are artificially peaked or uniformly flattened, the model reveals its underlying **miscalibration** while potentially still yielding **physically plausible** generations.

This thought experiment exposes a critical gap in how we evaluate video world models. At macroscopic scales, physical dynamics are effectively deterministic; apparent randomness arises from a sensitive dependence on initial conditions that cannot be fully resolved under sensory constraints (Strogatz, 2024). For a video model, the initial conditioning frame is inherently low-resolution and quantised, failing to capture the

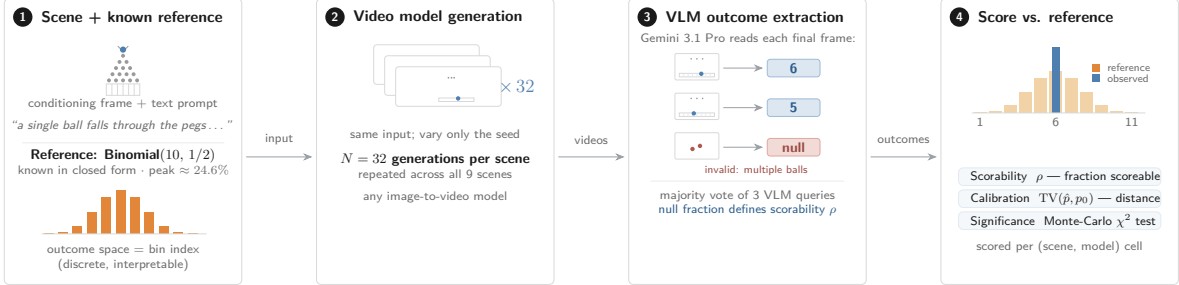

Figure 1: Overview of the CaliBench protocol. **(1)** Each scene pairs a conditioning frame and text prompt with an analytically known reference distribution over a discrete, interpretable outcome space (e.g. Binomial(10, $^1/_2$) over Galton-board bins). **(2)** A video model generates $N$=32 videos from identical inputs, varying only the random seed. **(3)** A vision–language model (Gemini 3.1 Pro, majority vote of three queries) extracts each video's discrete outcome from its final frame. **(4)** We compare the empirical outcome distribution to the reference along two orthogonal axes: scorability $\rho$ (fraction of scoreable generations) and calibration (total variation distance $\mathrm{TV}(\hat{p}, p_0)$ from the reference).

micro-states which dictate chaotic trajectories. To simulate these physics correctly, the model must map a single deterministic macro-state input to a distribution of diverse, valid trajectories via generative sampling.

A model is calibrated if its predicted distribution over outcomes matches the true distribution of those outcomes. The true calibration target is thus the marginal distribution of physical outcomes, integrated over the unresolved microscopic variations. For example, while the exact path of a ball through a peg array cannot be deterministically inferred from a starting frame, a physically grounded world model must reproduce the correct binomial distribution across landing bins. A model that collapses its generations entirely onto the central bin has failed to simulate the physics; it has erroneously learned an average behaviour rather than the distribution it averages over.

Existing generative video evaluation frameworks predominantly operate by comparing video samples within abstract, learned embedding spaces, *e.g.*, Fréchet Video Distance (FVD) or by assessing qualitative per-sample physical plausibility. While effective for auditing coarse perceptual quality, these methodologies yield uninterpretable, scalar-valued distances whose latent coordinates do not map to physical conservation laws, and they fundamentally ignore the dimension of statistical calibration.

To bridge this gap, we introduce **CaliBench** (Figure 1), an evaluation benchmark that profiles generative world models directly within discrete, low-dimensional, and physically meaningful outcome spaces, such as a Galton board bin index, a die face, or a roulette colour pocket. Because these environments encapsulate well-characterised random phenomena, their target reference distributions are analytically tractable and known in closed form, *e.g.*, uniform for fair dice, binomial for ideal Galton boards. Consequently, CaliBench enables the exact computation of distributional divergences against an absolute analytical ground truth, bypassing the need for biased empirical dataset proxies.

Our evaluation protocol decomposes world model performance into two orthogonal axes of failure: (i) **Scorability**: The fraction of video generations that produce a physically interpretable, completed outcome (e.g., a ball cleanly settling in a single bin without dissolving into motion blur, structural deformation, or artifacting); (ii) **Calibration**: The Total Variation Distance (TVD) between the empirical distribution of scoreable outcomes and the analytical reference distribution.

These metrics diagnose distinct model deficiencies. A model can exhibit low scorability but high calibration (generating few valid videos, but distributing them accurately), or high scorability alongside severe mode-collapse. Both dimensions are necessary to achieve a faithful physical simulation. We apply this protocol across nine distinct scenes spanning five distribution families where the ground-truth distribution is analytically known, visual outcomes are unambiguous, and bin assignments can be automatically extracted using a vision-language model (VLM).

**Contributions.**

1. **CaliBench**: We introduce a novel benchmark comprising nine physical simulation environments where outcomes map to interpretable, discrete spaces (e.g., bin indices, dice faces, suit/colour identifiers). By design, these environments possess analytically known reference distributions (binomial, Bernoulli, uniform, or skewed), enabling exact distributional calibration testing without relying on empirical dataset proxies or uninterpretable learned feature spaces.

2. **Orthogonal Evaluation Protocol**: We disentangle generative performance into two diagnostic axes: scorability ($\rho$), which quantifies the proportion of structurally valid generations, and calibration (Total Variation Distance), which measures distributional fidelity. We demonstrate that this decomposition is essential to isolate distinct failure modes—specifically, distinguishing between structural instability (low $\rho$) and mode collapse (high $\rho$ but high TVD)—which are otherwise conflated by aggregate accuracy metrics.

3. **Systematic Benchmarking**: We conduct the first large-scale calibration study of six state-of-the-art image-to-video models across these nine scenes. Our results utilise $\chi^2$ significance testing to demonstrate that most contemporary models are significantly miscalibrated, often collapsing probability mass despite producing individually "plausible" frames. We additionally probe (Appendix J) how classifier-free guidance affects this behaviour on the one model that exposes it. As our $\chi^2$ test takes calibration as the null hypothesis, it can only evidence miscalibration, never calibration; non-significant cells are inconclusive rather than calibrated. In addition, at $N{=}32$ generations per cell the test reliably detects only large miscalibrations (per-scene detectability floors in Appendix C).

4. **Reproducibility and Standardisation**: We release our comprehensive evaluation protocol, including the curated conditioning imagery, standardised VLM extraction prompts, and analysis codebase. We introduce the Mean Normalised Total Variation (mnTV) as a unified metric to track model progress toward physically grounded stochastic simulation.

## 2 Related Work

**Per-Sample Quality and Physical Plausibility.** Existing evaluation suites primarily focus on assessing individual video quality or per-sample physical adherence. VBench (Huang et al., 2024; Zheng et al., 2025) evaluates video rollouts along 16-plus dimensions; while it includes a diversity metric to detect near-identical generations, sample diversity is fundamentally distinct from calibration, as a diverse model can still be heavily mis-distributed. Other frameworks scrutinise specific structural properties of individual samples: FVMD (Liu et al., 2024) measures per-video motion consistency, the World Consistency Score (WCS) (Rakheja et al., 2025) tests causal compliance, WBench (Ying et al., 2026) evaluates interactive world models across physics and consistency axes, and PhysicsIQ (Motamed et al., 2025) tracks physical trajectories against specific ground-truth continuations. Similarly, automated auditing frameworks like PhyGenBench (Meng et al., 2024) leverage vision-language models (VLMs) to judge per-prompt physical common sense. These metrics are entirely orthogonal to our framework; they evaluate whether *each individual* video is internally consistent, whereas we interrogate whether the *ensemble of video generations* accurately recovers a known physical distribution.

**Feature-Space Distributional Metrics.** Standard distributional evaluation in generative video relies on metrics like FVD (Unterthiner et al., 2019) and JEDi (Luo et al., 2025). While these approaches capture appearance-level divergence by computing distances within learned perceptual feature spaces, they aggregate statistics across an entire dataset. This global aggregation renders them ill-suited for physical interpretability, as they cannot isolate the specific aleatoric uncertainty of a given scene. In contrast, our evaluation stratifies by scene, testing whether a specific analytical distribution is faithfully reproduced under a tightly controlled conditioning input.

**Distributional Calibration without a Known Reference.** A parallel line of work addresses image generation calibration against empirical datasets where no closed-form distribution is available. These methods typically implement two-sample tests within learned feature spaces, *e.g.*, by partitioning sample space into Voronoi cells (Richardson & Weiss, 2018), computing precision and recall on density estimates (Sajjadi et al.,

2018; Kynkäänniemi et al., 2019), or bounding the generator's effective support size via birthday paradox estimators (Arora et al., 2018). We side-step the inherent ambiguities of feature-space representations and empirical reference approximations. By designing physics experiments that project onto highly interpretable, discrete outcome spaces, we leverage precise, analytically known reference distributions.

**Uncertainty Estimation in Video Generation.** Quantifying uncertainty in video synthesis remains a nascent area. S-QUBED (Mei et al., 2025a) constructs predictive models to separate aleatoric and epistemic uncertainty in video generation, while $C^3$ (Mei et al., 2025b) generates per-pixel confidence maps for action-conditioned robotics models. However, neither framework evaluates model calibration against established physical distributions. CaliBench explicitly addresses this gap by auditing how accurately generative probabilities match physical reality

**Benchmarking Video World Models.** While recent literature increasingly frames video models as physical simulators, existing benchmarks remain confined to assessing per-sample physical plausibility. For instance, WorldModelBench (Li et al., 2025) assesses per-sample physics adherence; STEVO-Bench (Ma et al., 2026) evaluates the object permanence of occluded structures; TiViBench (Chen et al., 2025) evaluates per-sample reasoning; and PAI-Bench (Zhou et al., 2025) scores visual plausibility across video understanding tasks. Similarly, PhysicsIQ (Motamed et al., 2025) measures whether a generated continuation matches a single ground-truth physical trajectory, while PhyGenBench (Meng et al., 2024) leverages a VLM judge to evaluate per-prompt adherence to physical commonsense laws. All of these frameworks evaluate physical fidelity strictly at the per-sample level—checking whether *each individual* video is internally coherent. Although Yue et al. (2025) identify stochastic calibration as a defining capability for next-generation video world models, they do not formalise an experimental measurement protocol. **CaliBench** addresses this critical blind spot by testing the complementary distributional question: whether an *ensemble of video generations* accurately recovers a known physical distribution. A model can successfully pass existing per-sample checks by producing **plausible** individual videos, yet be **miscalibrated** and completely fail our population-level audit due to systemic mode collapse.

## 3 CaliBench

To evaluate models on their physical plausibility as well as their distributional calibration, CaliBench evaluates generative video world models across nine distinct statistical, physical scenes under a tightly controlled stochastic generation protocol.

### 3.1 Scene Definition

Each evaluation environment is formalised as a paired conditioning input $(x_0, c)$, where $x_0$ denotes the initial frame and $c$ represents the accompanying natural language prompt. For every model-scene pair, we sample an ensemble of $N = 32$ video trajectories to evaluate the empirical distribution of discrete macroscopic outcomes. The nine scenes are categorised into four distinct groups based on the analytical properties of their underlying ground-truth reference distributions (Figure 2).

**Binomial — Galton Boards.**

A Galton board consists of a triangular lattice of deflecting pegs. A falling ball hits each peg and undergoes a sequence of independent Bernoulli trials, deflecting left or right with equal probability. For a lattice with $n$ rows and $n+1$ landing bins, the probability $P(k)$ of the ball terminating in bin $k$ follows the binomial mass function:

$$P(k) = \binom{n}{k-1}\left(\tfrac{1}{2}\right)^n, \quad k = 1, \ldots, n+1. \tag{1}$$

We instantiate two variants with $n = 10$ rows (resulting in 11 discrete bins with a central mode at bin 6, where $P(6) \approx 24.6\%$): an *animated board* utilising a 2D vector graphic illustration with a green ball, and a photorealistic *physical board* captured via high-fidelity imagery with a metallic ball. While both setups share an identical analytical target distribution, their visual domains differ significantly, allowing us to evaluate whether a model's calibration is robust to low-level conditioning styles.

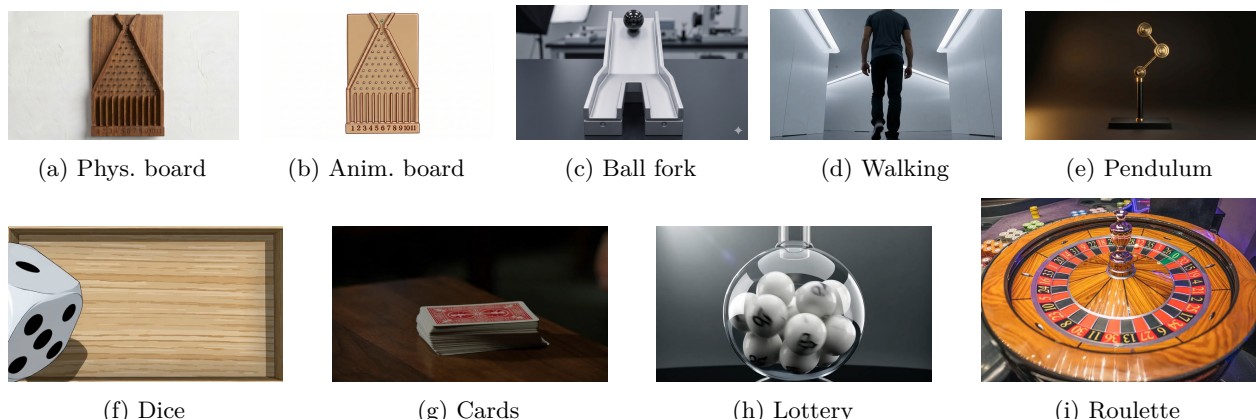

(a) Phys. board     (b) Anim. board     (c) Ball fork     (d) Walking     (e) Pendulum

(f) Dice     (g) Cards     (h) Lottery     (i) Roulette

Figure 2: Conditioning frames for all nine scenes. *Top row (left to right):* physical board, animated board [Binomial(10, 1/2)]; ball fork, walking, double pendulum [Bernoulli(1/2)]. *Bottom row:* dice, cards, lottery [Uniform]; roulette colour [known non-uniform].

**Bernoulli(1/2) — Symmetric Binary Environments.**

Three scenes isolate binary symmetric outcomes with equiprobable distributions ($P = 0.5$): (i) *Ball fork:* A ball rolls down a symmetric, bifurcated Y-shaped ramp, exiting via either the left or right channel; (ii) *Walking:* A human subject approaches a symmetric T-junction and executes either a left or right turn; (iii) *Double pendulum:* The horizontal orientation (left or right of the central vertical axis) of the outer mass is recorded at the final frame of the video. The latter frequently induces structural artefacts; rollouts are flagged as invalid if the model deforms the rigid, dual-rod mechanical geometry.

**Uniform Discrete — Dice, Cards, and Lottery.**

Three scenes evaluate uniform calibration over a discrete sample space of cardinality $k$: (i) *Dice:* A standard six-sided die is tossed into a bounded wooden box ($k = 6$); (ii) *Cards:* The top card is drawn from a face-down deck and scored strictly by its suit ($k = 4$); (iii) *Lottery:* A numbered ball is randomly ejected from a rotating tumbler into a transparent collection tube ($k = 20$). Cumulatively, these three setups probe calibration capabilities across scaling cardinalities.

**Known non-uniform — Roulette.**

We model an asymmetric multi-outcome distribution using a standard European roulette wheel. Rollouts are scored based on the final colour pocket where the ball settles, yielding an analytical reference distribution of $P(\text{red}) = P(\text{black}) = 18/37$ and $P(\text{green}) = 1/37$. This environment evaluates whether a model can accurately synthesise a known, skewed categorical distribution containing rare events. The conditioning prompt explicitly mandates that the ball must settle into a single numbered pocket; rollouts where the ball remains in motion or fails to settle represent a failure of text-prompt alignment and are categorised as unscorable.

**Reference Distributions and Dynamical Regime.**

Two of our scenes, the double pendulum and roulette, invoke reference distributions that presuppose chaotic operation. The double pendulum's marginal spatial distribution approaches uniformity only at energy thresholds sufficient to enter chaotic trajectories (Levien & Tan, 1993; Shinbrot et al., 1992). Within this regime, the system's short Lyapunov time ensures that unresolved microscopic variations in the initial frames are macroscopically amplified over the course of a multi-second video sequence. Similarly, the roulette reference distribution assumes multiple erratic deflections off the boundary frets, which requires high initial velocity. We induce these chaotic operational regimes directly through the initial conditioning frame configurations and text prompts (*e.g.,* starting the pendulum from an elevated potential energy state or appending "spinning rapidly" to the roulette prompt), though we do not explicitly compute the exact underlying energy thresholds.

Table 1: Per-model video-generation settings. All models receive the same conditioning frames, text prompts, and 32 fixed seeds; only the settings below differ. *Italic* marks a value equal to the provider's API default (whether we set it explicitly or left it unset); N/A means the model's API does not expose that parameter. Per-model durations are the shortest length each provider offers; WAN-2.7 (3 s) and SeeDance-2.0 (4 s) accept an arbitrary integer duration and were fixed slightly above the API minimum.

| Model | ID | Duration | Resolution | Prompt exp. | Neg. prompt | Audio |
|---|---|---|---|---|---|---|
| WAN-2.7 | `wan-video/wan-2.7-i2v` | 3 s | 720p | *yes* | *""* | N/A |
| SeeDance-2.0 | `bytedance/seedance-2.0` | 4 s | 480p | N/A | N/A | off |
| HappyHorse-1.0 | `alibaba/happyhorse-1.0` | 3 s | 720p | N/A | N/A | N/A |
| Veo 3.1 | `google/veo-3.1` | 4 s | 720p | N/A | *unset* | *on* |
| Runway Gen-4.5 | `runwayml/gen-4.5` | *5 s* | N/A | N/A | N/A | N/A |
| Cosmos3-Super | nvidia/Cosmos3-Super (local) | 5.04 s[†] | 1280×704 | N/A | *set* | N/A |

[†]Cosmos3-Super: 121 frames @ 24 fps; `steps`=35, `guidance_scale`=6.0.

Chaotic operation alone, however, does not guarantee the symmetric Bernoulli($1/2$) marginal we assign to the pendulum: a chaotic trajectory could still be biased towards a particular side of the vertical axis at a particular time. Therefore we verify the Bernoulli($1/2$) marginal for our system by simulation. Integrating the full nonlinear double-pendulum equations of motion from the conditioning frame's measured release state ($\theta_1 \approx 146°$, $\theta_2 \approx 215°$, length ratio $l_2/l_1 = 0.82$, equal masses, released from rest), we propagate a dense ensemble of micro-perturbations (independent Gaussian noise of standard deviation $0.5°$ on each release angle) about that state—the sub-frame micro-state a video model cannot resolve from a single frame—with an adaptive RK45 integrator. The simulated marginal $P$(left) of the outer mass converges towards 0.50 as the trajectory mixes, staying within 0.05 of 0.50 across the range of model video lengths ($t$ from 3 to 5 s); below roughly one second the outcome is still near-deterministic and has not yet mixed. The same construction—selecting a release configuration and reading off its simulated marginal—could equally yield scenes with analytically characterised but deliberately *skewed* references; we do not pursue this here, but note it as an exciting avenue for future work.

## 3.2 Generation Protocol

We evaluate six state-of-the-art image-to-video architectures: WAN-2.7 (Wan Team, Alibaba, 2025), SeeDance-2.0 (ByteDance Seed, 2026), HappyHorse-1.0 (Alibaba Group, 2026), Veo 3.1 (Google DeepMind, 2025), Runway Gen-4.5 (Runway, 2025), Cosmos3-Super (Agarwal et al., 2026).

For each scene–model combination, we produce 32 distinct video generations from identical inputs modulated only by the integer random seed passed to the respective model APIs, drawn from a fixed set of 32 unique seed integers representing the source of stochastic variance. All other configuration variables—including the set of conditioning frames and text prompts (detailed in Appendix G), generation duration, and target output resolution are held static per model (Table 1). Default sampling parameters (*e.g.*, classifier-free guidance scales, internal prompt expansion, temperature, etc.) are maintained at the respective model defaults.

We deliberately choose an ensemble size of $N = 32$ generations to balance statistical validity with computational feasibility. Evaluating a single architecture across all nine scenes requires $9 \times 32 = 288$ video generations, keeping the computational overhead on par with related video physics benchmarks such as PhysicsIQ (Motamed et al., 2025) and PhyGenBench (Meng et al., 2024). Figure 3 provides illustrative frame strips comparing a valid, scorable rollout against a structural simulation failure (spurious multi-object generation).

## 3.3 Outcome Extraction

Discrete outcomes are programmatically extracted from the synthesised videos using Gemini 3.1 Pro (Google, 2026). We deploy scene-specific prompts that instruct the vision-language model (VLM) to map the visual trajectory to a natural language token matching the target outcome space (*e.g.*, "left", "right", an explicit bin index, a suit, or a colour pocket). If the terminal state is structurally ambiguous or unresolvable, the VLM is

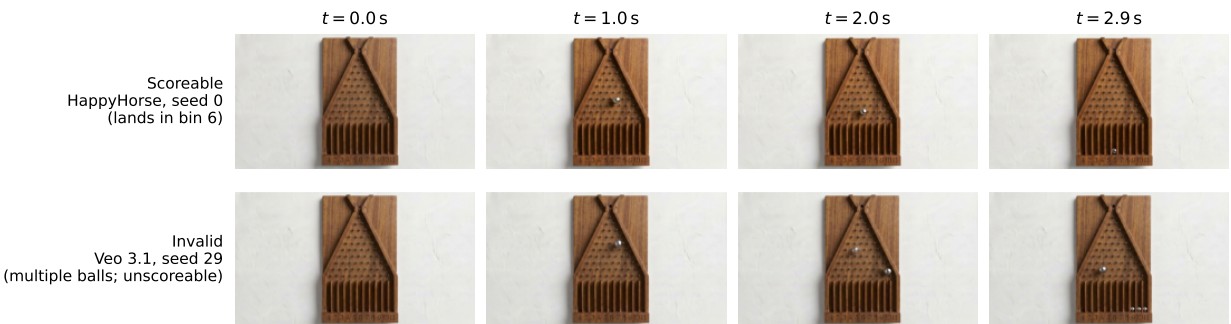

Figure 3: Frame strips from two physical Galton-board video generations illustrating the two outcome categories. *Top:* a scoreable video generation (HappyHorse): a single ball is released, falls through the peg array, and comes to rest in a single bin. *Bottom:* an invalid video generation (Veo 3.1): the model generates multiple balls during the simulation, leaving the outcome unscoreable. Failures of this type drive the scorability $\rho$ metric below 1.

explicitly directed to return a `null` value. The comprehensive prompt suite is provided in Appendix H. The `null` assignment reflects failure modes observed during prompt development:

- *Roulette:* The ball must visually come to rest entirely within a single pocket boundary, if the ball disappears, floats outside the track, or balances statically on an intersecting fret are marked `null`.

- *Lottery:* Exactly one distinct ball must occupy the transparent exit tube.

- *Dice:* A single, identifiable die face must be oriented parallel to the upper perspective plane.

- *Cards:* The top card must be visually drawn and resolved as a standard, recognizable suit.

- *Double Pendulum:* The structural integrity of the rigid dual-rod mechanism must be preserved without topological fractures or geometric morphing throughout the video duration.

To reduce variance from VLM imprecision, we run each evaluation query three times at the model's default sampling temperature and aggregate by majority vote (ties resolved to `null`). A video generation is classified as **valid** if the majority vote returns an unambiguous outcome and **invalid** otherwise.

We validate the extraction pipeline by comparing against human annotation on a stratified sample of 18 video generations per scene (162 videos total), achieving $152/162 = 93.8\%$ overall agreement with a low $3.7\%$ false-null rate. Appendix F reports the per-scene breakdown.

### 3.4 Metrics

We evaluate each model along two orthogonal axes (Section 3.4.1), test the statistical significance of the resulting metrics, and aggregate them into a single score via the algorithm in Section 3.4.2.

### 3.4.1 Evaluation Axes

To comprehensively evaluate video world models, we decouple performance into two orthogonal axes: **scorability ($\rho$)** and **(Total Variation Distance)**.

**Scorability $\rho$.** The scorability metric $\rho$ quantifies the proportion of scorable generations within an ensemble:

$$\rho = \frac{n_{\text{valid}}}{N}, \tag{2}$$

where $N = 32$ denotes the total number of generated video rollouts. It is a necessary but insufficient condition for a faithful physical simulation. A model can achieve a perfect score of $\rho = 1$ while exhibiting severe

calibration failures. Furthermore, a scorable generation does not inherently guarantee that the generated video is physically plausible.

**Effect size: total variation distance (TV).** Let $\hat{p}_{\text{valid}}$ denote the empirical categorical distribution computed over the $n_{\text{valid}}$ successful video generations for a given scene–model combination. We quantify the calibration error as the Total Variation Distance (TVD) between the empirical distribution $\hat{p}_{\text{valid}}$ and the analytical reference distribution $p_0$:

$$\text{TV}(\hat{p}_{\text{valid}}, p_0) = \tfrac{1}{2} \sum_{i=1}^{k} |\hat{p}_{\text{valid},i} - p_{0,i}| \in [0, 1], \tag{3}$$

reading as the fraction of probability mass misallocated (Levin & Peres, 2017). We select TVD rather than an Earth Mover's (Wasserstein) Distance because many evaluation scenes feature unordered, categorical sample spaces, such as card suits or roulette colours, where an inter-outcome distance metric cannot be naturally defined. Although TV ignores the ordinal spatial proximity present in the Galton board, it provides a unified, cross-scene metric that safely accommodates our purely nominal categories. At finite sample sizes, *e.g.*, $n_{\text{valid}}$ generations, the plug-in estimator for TVD is inherently biased upward: a perfectly calibrated model produces non-zero TV from sampling noise alone, and this null floor grows with the number of outcomes $k$. We do not correct TV per scene–model combination. Therefore, raw per-sample TV values are comparable *within* one scene only and not across scenes. The benchmark-aggregate score below makes the cross-scene comparison explicit via floor subtraction. Because this estimate is evaluated over the subset of scorable generations, it represents the outcome distribution *conditional on validity*, a framework aligning with the true target distribution under a Missing Completely At Random (MCAR) assumption (see Section 5 for an extended justification).

### 3.4.2 Statistical Inference and Aggregation

**Significance.** While TV provides an interpretable effect size, verifying whether an observed divergence is statistically distinguishable from pure sampling noise requires an independent inferential framework. We evaluate the null hypothesis $H_0$, that the generated outcomes are sampled directly from the true physical reference distribution, using a Monte Carlo exact $\chi^2$ test applied to the valid rollout subset. We deliberately treat both as distinct objects: TV describes the size of the miscalibration and never carries a significance marker, while the $\chi^2$ test is reported as a verdict (and a $p$-value) and is never read as a magnitude in the main text. The $\chi^2$ test exhibits higher sensitivity than TV to deviations on low-probability bins, *e.g.*, the rare green roulette pocket. Hence, the two quantities are complementary rather than redundant.

For each scene–model combination we compute Pearson's $\chi^2$ statistic on the valid generations against the reference distribution:

$$\chi^2 = \sum_{i=1}^{k} \frac{(O_i - E_i)^2}{E_i}, \tag{4}$$

where $O_i$ and $E_i$ represent the observed and expected counts, respectively. We obtain the $p$-value via Monte Carlo: 50,000 independent samples of size $n_{\text{valid}}$ are drawn from the reference distribution and the $p$-value is defined as the fraction of replicates whose $\chi^2$ statistic equals or exceeds the observed value. Utilising an asymptotic $\chi^2_{k-1}$ reference would be unreliable at $N = 32$ because Cochran's expected-count rule is violated on four scenes (Cochran, 1954). Specifically, the Galton boards yield $E_i < 1$ in the tail bins, the green roulette bin yields $E_{\text{green}} \approx 0.86$, and the uniform lottery bounds all bins at $E_i \approx 1.6$. Constructing an exact empirical distribution via Monte Carlo (Hope, 1968) circumvents that issue. Each compound Monte Carlo $p$-value is the add-one estimator $\hat{p} = (b+1)/(M+1)$ of Hope (1968), a binomial proportion over $M = 50,000$ i.i.d. draws under $H_0$. Its corresponding standard error is formalised as:

$$\text{SE}(\hat{p}) = \sqrt{\frac{p(1-p)}{M}}. \tag{5}$$

It is bounded at its maximum $p = 0.5$, where $\text{SE} = \sqrt{0.25/50{,}000} \approx 0.0022$, and falls to $\sqrt{0.0475/50{,}000} \approx 0.00097$ near $p = 0.05$. Therefore, it resolves $p$-values far more finely than the $\alpha = 0.05$ decision threshold requires.

Because each of the scene–model cells is tested separately, we additionally control the false discovery rate across the family of testable cells using the Benjamini–Hochberg procedure at $\alpha = 0.05$ (Benjamini & Hochberg, 1995). The verdicts for each scene–model combination are reported after this correction, and Appendix D lists the raw, Benjamini–Hochberg, and Bonferroni $p$-values for every cell.

**Benchmark score: floor-referenced aggregate.** To establish an aggregate leaderboard ranking, the per-scene–model TV does not suffice: TV alone neither goes to zero at calibration nor mixes commensurably across scenes of different $k$. To resolve this, we define a per-scene–model excess by subtracting the expected sampling noise floor:

$$e_{\mathrm{sm}} = \frac{\mathrm{TVD} - \mathbb{E}_{H_0}[\mathrm{TVD}]}{1 - \mathbb{E}_{H_0}[\mathrm{TVD}]}, \tag{6}$$

where the per-scene–model null floor $\mathbb{E}_{H_0}[\mathrm{TV}]$ has a closed form derived in Appendix A. The floor is therefore a deterministic function of $(p_0, n_{\mathrm{valid}})$ that the released code evaluates at each scene–model's $n_{\mathrm{valid}}$. We define the per-scene–model score as $s_{\mathrm{sm}}$:

$$s_{\mathrm{sm}} = (1 - \rho) + \rho \cdot e_{\mathrm{sm}}. \tag{7}$$

Our final benchmark metric *mean normalised Total Variation* (**mnTV**) is given by the mean of $s_{\mathrm{sm}}$ across all nine scenes, bounded above by 1, and approaches zero when every scene is both fully scoreable and calibrated up to sampling noise. We compute bootstrap confidence intervals for mnTV to enable fair comparison of models. We emphasise that mnTV is a one-number summary intended to sit alongside the per-scene grid, not in place of it: averaging over heterogeneous scenes masks per-scene reversals, *e.g.*, a model that is best on the ball fork can be most collapsed on lottery, and the score conflates the two axes (scorability and calibration) that the rest of the paper treats as orthogonal.

A model with mnTV near zero would be both highly scorable *and* calibrated — producing scoreable videos whose outcome distribution matches the reference up to sampling noise.

## 4 Results

We evaluate six frontier architectures on **CaliBench**. Table 2 reports the per-scene–model scorability $\rho$ and Total Variation Distance (TVD) across all nine environments as well as the aggregate Mean Normalised Total Variation (mnTV). While SeeDance-2.0 yields the lowest overall calibration penalty (mnTV = 0.39), overlapping bootstrap intervals indicate that models resolve into broad performance tiers rather than a strict ranking. Crucially, all empirical aggregates sit far above the baseline of a perfectly calibrated operator (mnTV $\approx$ 0). Table 3 reports $p$-values from our Monte Carlo $\chi^2$ test. While TVD measures how far the empirical distribution sits from the reference, $p < 0.05$ in the $\chi^2$ test means that sampling noise alone is an implausible explanation for that gap at a 95% confidence level. These metrics are complementary: because the $\chi^2$ statistic normalises deviations by expected counts, it exhibits high sensitivity to anomalies in low-probability bins that contribute minimally to the absolute TVD sum. Neither the differing output resolutions nor video durations account for these deficits: regenerating SeeDance-2.0 at 720p, and re-running the applicable models at a uniform 5 s, both leave the calibration results essentially unchanged (Appendix I).

The main finding is heterogeneity: across the eight informative scenes no model dominates, and the per-scene best swaps between models (Runway Gen-4.5 leads on the animated Galton board, SeeDance-2.0 on the physical board and lottery, HappyHorse on the ball fork, Cosmos3-Super on walking and dice, and Veo 3.1 on roulette; cards is a three-way tie in TV). Despite these localised strengths, within most scenes every model is significantly miscalibrated; the few unrejected cells lie inside or close to their null band and should be read together with each scene's detectability floor (Appendix C).

Applying the multiple-comparison correction (Section 3.4.2) leaves 28 of the 44 testable cells significant. The severe mode collapses ($p < 0.001$) survive any correction, including the far stricter Bonferroni bound ($p \leq 0.00114$); two cells change verdict relative to an uncorrected threshold. Runway Gen-4.5 ($p = 0.038 \rightarrow q = 0.057$) on the physical board and SeeDance-2.0 on lottery ($p = 0.049 \rightarrow q = 0.070$) are no longer rejected, joining the small set of cells whose deviation from calibration is indistinguishable from sampling noise. Borderline cells elsewhere are unaffected — Cosmos3-Super on the dice, for instance, remains significant after correction ($p = 0.011 \rightarrow q = 0.019$). The full raw and adjusted $p$-values are in Appendix D.

Table 2: **Per-scene–model descriptive results across all nine evaluation environments.** We evaluate the Total Variation Distance TV from the scene's reference distribution on the valid sample (lower is better, range $[0, 1]$). †: small $n_{valid}$ ($\leq 3$), so per-scene–model distributional statements are unreliable. —: 0 valid video generations. Bold: lowest TV per scene among non-† cells. mnTV footer: $[2.5\%, 97.5\%]$ bracketed values are the bootstrap interval from 50,000 resamples within each scene–model combination (Appendix A). Mean Normalised Total Variation (mnTV): mean of $s_{sm}$ across all nine scenes, scene–model decomposition in Appendix A. For reference, a perfectly calibrated, fully scoreable model has mnTV $\approx 0$ (central-95% range $\approx [-0.04, 0.04]$ over 50,000 synthetic draws); the observed mnTVs in the footer all sit far above this null range. Pendulum is daggered for all six models — none yields more than three valid generations (Cosmos3-Super three, SeeDance-2.0 two, the rest at most one): a universal structural failure.

| | WAN-2.7 | | SeeDance-2.0 | | HappyHorse | | Veo 3.1 | | Runway 4.5 | | Cosmos3-Super | |
|---|---|---|---|---|---|---|---|---|---|---|---|---|
| Scene | $\rho\uparrow$ | TV$\downarrow$ | $\rho\uparrow$ | TV$\downarrow$ | $\rho\uparrow$ | TV$\downarrow$ | $\rho\uparrow$ | TV$\downarrow$ | $\rho\uparrow$ | TV$\downarrow$ | $\rho\uparrow$ | TV$\downarrow$ |
| *Binomial*(10, 1/2) — *11 bins* | | | | | | | | | | | | |
| Physical board | 0.69 | 0.71 | 0.59 | **0.27** | 0.88 | 0.58 | 0.03 | 0.99† | 0.56 | 0.37 | 0.19 | 0.50 |
| Animated board | 0.91 | 0.75 | 0.88 | 0.66 | 0.91 | 0.68 | 0.97 | 0.49 | 0.88 | **0.20** | 0.03 | 0.99† |
| *Bernoulli*(1/2) — *2 outcomes* | | | | | | | | | | | | |
| Ball fork | 1.00 | 0.25 | 1.00 | 0.19 | 0.94 | **0.00** | 0.97 | 0.24 | 0.91 | 0.12 | 0.84 | 0.06 |
| Walking | 0.97 | 0.21 | 1.00 | 0.19 | 1.00 | 0.12 | 1.00 | 0.50 | 1.00 | 0.03 | 0.84 | **0.02** |
| Pendulum | 0.00 | — | 0.06 | 0.00† | 0.00 | — | 0.03 | 0.50† | 0.00 | — | 0.09 | 0.50† |
| *Uniform*$\{1, \ldots, k\}$ | | | | | | | | | | | | |
| Dice ($k=6$) | 1.00 | 0.68 | 1.00 | 0.42 | 0.97 | 0.80 | 1.00 | 0.83 | 0.94 | 0.40 | 1.00 | **0.28** |
| Cards ($k=4$) | 0.56 | 0.58 | 0.91 | **0.50** | 0.38 | **0.50** | 0.34 | **0.50** | 0.03 | 0.75† | 0.03 | 0.75† |
| Lottery ($k=20$) | 0.88 | 0.69 | 0.75 | **0.40** | 0.78 | 0.95 | 0.59 | 0.90 | 0.62 | 0.50 | 0.22 | 0.70 |
| *Known non-uniform:* $P(red) = P(black) = {}^{18}/_{37}$, $P(green) = {}^1/_{37}$ | | | | | | | | | | | | |
| Roulette | 0.69 | 0.26 | 0.84 | 0.12 | 0.97 | 0.19 | 0.66 | **0.11** | 0.22 | 0.12 | 0.38 | 0.47 |
| **mnTV**$\downarrow$ | 0.58 [0.54,0.63] | | **0.39** [0.38,0.48] | | 0.54 [0.51,0.59] | | 0.65 [0.63,0.69] | | 0.48 [0.48,0.57] | | 0.63 [0.62,0.70] | |

Table 3: Monte Carlo $p$-values from the $\chi^2$ test against the reference distribution (50,000 null hypothesis ($H_0$) replicates per scene–model combination, seed $= 0$). $H_0$: outputs are i.i.d. from the reference (calibrated). Values shown are *raw* $p$-values; we control the false discovery rate across the 44 computed tests with the Benjamini–Hochberg procedure at $\alpha = 0.05$ (BH critical threshold $p \leq 0.0297$; full raw and adjusted values in Appendix D). †: $\leq 3$ valid video generations; —: 0 valid video generations. ‡: significant at raw $\alpha = 0.05$ but *not* after Benjamini–Hochberg correction ($q > 0.05$).

| Scene | WAN-2.7 | SeeDance | HappyHorse | Veo 3.1 | Runway | Cosmos3-Super |
|---|---|---|---|---|---|---|
| *Binomial*(10, 1/2) | | | | | | |
| Physical board | 0.005 | 0.088 | 0.004 | † | 0.038‡ | 0.013 |
| Animated board | <0.001 | <0.001 | 0.001 | 0.057 | 0.002 | † |
| *Bernoulli*(1/2) | | | | | | |
| Ball fork | 0.007 | 0.051 | 1.000 | 0.011 | 0.264 | 0.699 |
| Walking | 0.0297 | 0.051 | 0.214 | <0.001 | 0.859 | 1.000 |
| Pendulum | — | † | — | † | — | † |
| *Uniform discrete* | | | | | | |
| Dice ($k=6$) | <0.001 | <0.001 | <0.001 | <0.001 | <0.001 | 0.011 |
| Cards ($k=4$) | <0.001 | <0.001 | 0.008 | <0.001 | † | † |
| Lottery ($k=20$) | <0.001 | 0.049‡ | <0.001 | <0.001 | 0.010 | 0.696 |
| *Known non-uniform* | | | | | | |
| Roulette | 0.004 | 0.004 | 0.096 | 0.490 | 0.279 | <0.001 |

To provide further intuition for the results presented in this section, Figure 4 shows empirical bin distributions for the Galton board scenes, overlaid on the reference binomial(10, 1/2) (for remaining distribution plots refer to Appendix B). Figure 5 maps the evaluation space by plotting scorability $\rho$ against TVD alongside a shaded envelope tracking the expected 95[th] percentile of null sampling noise. Because this visual boundary reflects unweighted TVD thresholds, it functions as a purely descriptive baseline. Consequently, individual cells with generative errors concentrated entirely on low-probability bins can trigger an inferential rejection via the $\chi^2$ test while visually remaining within the shaded geometric envelope.

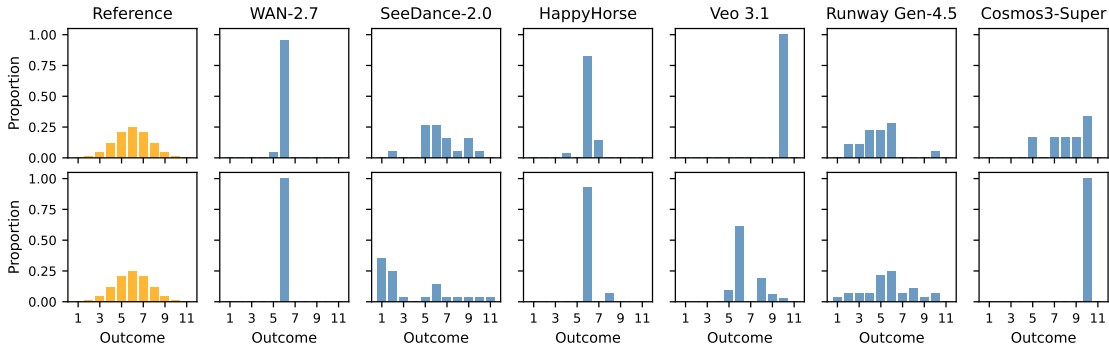

Figure 4: Empirical bin distributions for each model (blue) vs. ground-truth Binomial(10, 1/2) (orange). Top: physical board. Bottom: animated board. Most models over-concentrate near the central bin, far exceeding the expected peak probability of 24.6%; on the animated board SeeDance-2.0 instead places excessive probability mass on the left tail (bin 1 and 2).

## 5   Discussion, Limitations, and Future Work

**Discussion.** Our evaluation reveals that the dominant failure mode in frontier world models is systemic probability mass over-concentration, often manifesting as total mode collapse despite high individual sample realism. This behaviour stems from a mixture of shared, a priori biases (*e.g.*, over-representing the central modal bin of Galton boards) and localised conditioning cues present in the initial image (Appendix K). Crucially, performance fails to generalise across distribution families: models nearing perfect calibration on binary symmetric environments degrade sharply on high-cardinality or highly skewed targets, proving that physical calibration is governed by factors deeper than the mathematical structure of the target reference distribution. These findings expose severe risks for utilising world models to stress-test autonomous systems in safety-critical domains (*e.g.*, autonomous driving or robotics), where simulating the full variance of plausible futures is mandatory. Relying on current uncalibrated priors yields overly deterministic rollouts that mask critical tail risks. Mechanistically, classifier-free guidance (CFG) acts as a critical dial; our sweeps (Appendix J) show that lowering CFG strength mitigates over-concentration but directly trades away structural scorability ($\rho$), uncovering a fundamental optimisation tension between rendering validity and distributional calibration.

**Limitations.** Several limitations apply. We evaluate image-to-video pipelines exclusively, leaving text-to-video stochasticity for future work. Additionally, our automated VLM outcome extraction introduces a minor noise floor, though human validation verifies a 93.8% annotation agreement. Mechanistically, our TVD metric conditions on scorable rollouts; while localised rendering pathologies (*e.g.*, object merging) drive unscorability, validation suggests these failures are independent of the physical trajectories the dynamics resolve toward. Finally, CaliBench evaluates marginal outcome distributions rather than full continuous trajectory spaces. Because several evaluated models are closed-weight commercial systems, we cannot audit their training corpora for depictions of our scene families; we note, however, that memorised process statistics for an unbiased dataset would be expected to improve calibration on these canonical textbook processes.

**Future Work.** A natural path forward is integrating distributional verification directly into model pre-training or alignment phases (Yuan et al., 2026). While multi-sample evaluation is computationally intensive—requiring 1,728 video rollouts and 5,184 VLM queries across our current matrix—it remains highly viable for low-step or autoregressive architectures, providing a critical engineering pathway toward calibrated physical simulators. The protocol described in this paper could be used to create larger benchmarks in the future, with a larger number of reference scenes — we believe the current number of scenes provides a pragmatic balance between computational expense and evaluation fidelity. A natural extension is testing whether calibration on CaliBench's closed-form scenes predicts distributional fidelity on open-world footage, where reference distributions must instead be estimated empirically from large observational samples.

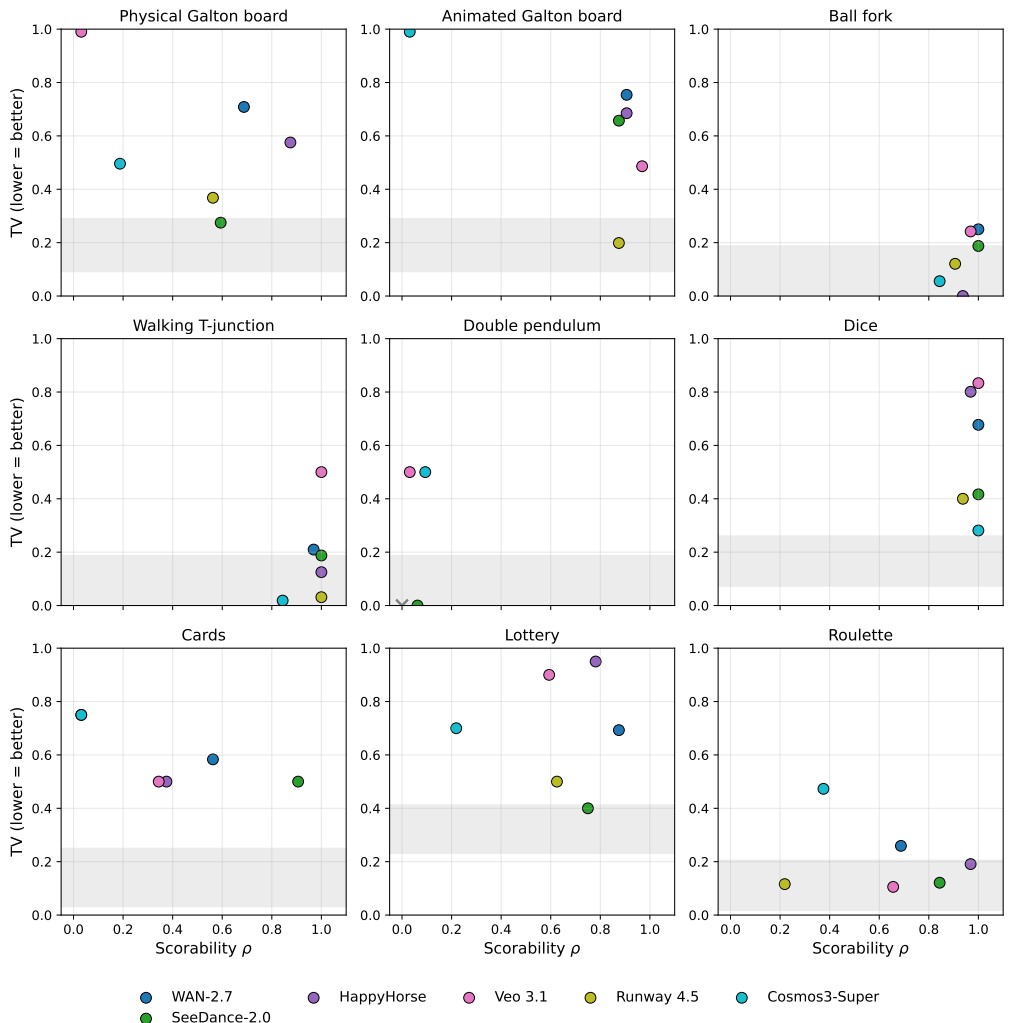

Figure 5: Scorability vs. total variation distance TV for all nine scenes. The bottom-right corner of each panel (high scorability, low TV) is the calibration target; no evaluated model approaches it consistently across scenes. The shaded band per panel is the central-95% envelope of TV under perfect calibration at $n_{\text{valid}} = N$, shown as a descriptive aid for interpreting the magnitude of TV relative to sampling noise — it is not a rejection region, and a marker's position relative to the band is not equivalent to the calibration verdict (which comes from the Monte Carlo exact $\chi^2$ test). The band is drawn at $n_{\text{valid}} = N = 32$ per panel and does not directly characterise sampling noise for cells with $\rho < 1$ (those cells have a wider envelope, not shown). Because of the upward bias of the TV estimator, the shaded null band does not necessarily include TV $= 0$.

# 6 Conclusion

In this work, we introduced **CaliBench**, an evaluation framework designed to benchmark the statistical calibration of generative video world models within discrete, physically meaningful outcome spaces. By evaluating six state-of-the-art models across nine classic stochastic environments, we demonstrated that current architectures routinely fail to function as reliable stochastic simulators. Even when generating highly realistic, physically plausible individual trajectories, these models systematically suffer from severe probability mass over-concentration and mode collapse. By formalising the distinct properties of **scorability** and **distributional calibration**, CaliBench exposes critical vulnerabilities in the stochastic dynamics of frontier world models, establishing a rigorous mathematical baseline necessary for developing reliable, safe, and truly calibrated physical simulation pipelines.

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

## A $\mathrm{mnTV}$ **Definition and Per-Cell Decomposition**

Our metric $\mathrm{mnTV}$ aggregates per-scene calibration into a single, noise-corrected scalar. The underlying scene–model components are defined as:

$$e_{\mathrm{sm}} = \frac{\mathrm{TV}(\hat{p}_{\mathrm{valid}}, p_0) - \mathbb{E}_{H_0}[\mathrm{TV}]}{1 - \mathbb{E}_{H_0}[\mathrm{TV}]}, \tag{8}$$

$$s_{\mathrm{sm}} = (1 - \rho) + \rho \cdot e_{\mathrm{sm}}, \tag{9}$$

where the expected null term $\mathbb{E}_{H_0}[\mathrm{TV}]$ (Equation 14) accounts for finite-sample noise. Consequently, $e_{\mathrm{sm}}$ isolates the true excess calibration error attributable to model bias, with $e_{\mathrm{sm}} = 0$ at perfect calibration (up to sampling noise) and unity representing total mass concentration on a single outcome that has zero probability under the reference distribution. Because the excess is signed, $e_{\mathrm{sm}}$ can be marginally negative for cells better-calibrated than the noise floor.

Unscorable generations are penalised via the $(1-\rho)$ term which reflects the maximum distributional divergence since the physical reference distribution places zero probability mass on the null outcome. Because this term dominates the score on the null-heavy scenes (cards, roulette, lottery, pendulum), it is possible that if the models' scorability were improved our conclusion about the models' calibration could significantly change. Therefore we show in Appendix E the effect of bounding each cell's TV under best and worst-case null imputation. The final benchmark scalar is computed as the arithmetic mean across all evaluated environments:

$$\mathrm{mnTV} = \frac{1}{N_{\mathrm{scenes}}} \sum_{c=1}^{N_{\mathrm{scenes}}} s_{\mathrm{sm}} \leq 1. \tag{10}$$

This formulation explicitly includes the chaotic double pendulum environment, which contributes approximately $1 - \rho$ for every model because the $\rho \, e_{\mathrm{sm}}$ term is negligible: no model produces more than three valid samples, too few to yield an informative $e_{\mathrm{sm}}$.

Crucially, $\mathrm{mnTV}$ values are inherently conditioned on the specific sample size $N$ and the chosen VLM outcome extraction protocol. To preserve direct baseline comparability with the leaderboard tracking established in this work, these experimental hyperparameters must remain invariant.

*We note that* $\mathrm{mnTV}$ *aggregates nine heterogeneous scenes characterised by distinct reference distributions and cardinalities; per-scene performance reversals are expected, i.e., a model achieving strong calibration in one environment may exhibit total mode collapse in another. Because the score conflates the two orthogonal axes of scorability and calibration, we report* $\mathrm{mnTV}$ *strictly as a convenient one-number ordering tool for benchmark users, rather than a substitute for the comprehensive per-scene evaluation grid detailed in Table 2.*

**Null** $\mathrm{mnTV}$ **Baseline.** To ground the empirical mnTV scores presented in Table 2, we establish an analytical baseline for an idealised, perfectly calibrated, and fully scorable operator. A Monte Carlo simulation of 50,000 such configurations—setting $\rho = 1$ across all environments and drawing $N = 32$ outcomes per scene–model cell i.i.d. from the true reference distribution $p_0$—yields a null distribution centred on zero, with mean $\approx 0$, median $\approx 0$, and a central 95% interval of $\approx [-0.04, 0.04]$. Because the excess term $e_{\mathrm{sm}}$ is signed and the null floor $\mathbb{E}_{H_0}[\mathrm{TV}]$ is computed exactly in closed form, $\mathrm{TV} - \mathbb{E}_{H_0}[\mathrm{TV}]$ is exactly mean-zero under calibration, so a perfectly calibrated model scores zero in expectation. All six evaluated frontier models exhibit an mnTV

that sits far above this null range. The lowest empirical error observed is SeeDance-2.0 at 0.39, well above the sampling-noise floor, confirming that the observed calibration deficits cannot be explained by finite-sample fluctuations. This null distribution serves as the reference baseline reported in the caption of Table 2. Table 4 reports the resulting per-scene–model scores $s_{\text{sm}}$, with the aggregate mnTV in the bottom row.

**Bootstrap Uncertainty Intervals.** The bracketed values reported in the mnTV row of Table 2 denote central 95% percentile intervals computed via 50,000 bootstrap iterations. Because the specific suite of nine scenes defines the benchmark itself rather than a random draw from an environment population, the scene composition is held static across resamples. Instead, the bootstrap scheme isolates finite-sample variance within each scene–model configuration by resampling all $N = 32$ video generations with replacement from the cell's empirical multinomial over the valid outcome bins plus a null (unscorable) category, so that both the scorability ratio $\rho$ and $n_{\text{valid}}$ vary across replications. For each bootstrap replication, we recompute $\rho$, TV, and the corresponding excess error $e_{\text{sm}}$ using the closed-form null floor $\mathbb{E}_{H_0}[\text{TV}]$ re-evaluated at the resampled $n_{\text{valid}}$. This uncertainty interval thus quantifies the sensitivity of mnTV purely to the finite allocation of $N = 32$ video generations per cell. We report these empirical quantiles directly.

Table 4: Per-scene–model score $s_{\text{sm}}$ (Eq. 9), with mnTV (mean across nine scenes) in the bottom row. Lower is better; bounded above by 1, with 0 the calibrated expectation (small negatives occur for cells better-calibrated than the sampling-noise floor). Bold: lowest per scene; mnTV: lowest overall.

| Scene | WAN-2.7 | SeeDance | HappyHorse | Veo 3.1 | Runway | Cosmos3-Super |
|---|---|---|---|---|---|---|
| *Binomial*$(10, 1/2)$ | | | | | | |
| Physical board | 0.75 | **0.44** | 0.54 | 1.00 | 0.54 | 0.84 |
| Animated board | 0.73 | 0.63 | 0.65 | 0.39 | **0.13** | 1.00 |
| *Bernoulli*$(1/2)$ | | | | | | |
| Ball fork | 0.19 | 0.13 | **-0.01** | 0.21 | 0.14 | 0.14 |
| Walking | 0.17 | 0.13 | 0.06 | 0.46 | **-0.04** | 0.10 |
| Pendulum | 1.00 | **0.92** | 1.00 | 0.97 | 1.00 | 0.94 |
| *Uniform discrete* | | | | | | |
| Dice ($k$=6) | 0.62 | 0.31 | 0.77 | 0.80 | 0.33 | **0.15** |
| Cards ($k$=4) | 0.72 | **0.48** | 0.77 | 0.78 | 0.97 | 0.97 |
| Lottery ($k$=20) | 0.59 | **0.30** | 0.94 | 0.90 | 0.51 | 0.78 |
| *Known non-uniform* | | | | | | |
| Roulette | 0.43 | 0.19 | **0.15** | 0.35 | 0.76 | 0.77 |
| mnTV$\downarrow$ | 0.58 | **0.39** | 0.54 | 0.65 | 0.48 | 0.63 |

**Null floor: a closed-form, deterministic function.**

The per-scene–model excess error $e_{\text{sm}}$ requires subtracting the expected null variation $\mathbb{E}_{H_0}[\text{TV}]$. This baseline depends exclusively on the scene's true reference distribution $p_0$ and the number of valid samples $n_{\text{valid}}$. Crucially, because a model's specific $n_{\text{valid}}$ is dynamically determined by the VLM's automated filtration process, this baseline cannot be pre-computed as a static scalar constant. Evaluating the baseline requires a flexible, closed-form analytic expression parameterised by the realised value of $n_{\text{valid}}$.

Let $X_i$ denote the number of valid video generations whose extracted outcomes fall within bin $i$, such that the empirical probability is $\hat{p}_{\text{valid},i} = X_i/n_{\text{valid}}$. Substituting this expression into the definition of Total Variation Distance (TV) and factoring the scalar sample count out of the absolute value yields:

$$\text{TV}(\hat{p}_{\text{valid}}, p_0) = \tfrac{1}{2} \sum_i \left| \tfrac{X_i}{n_{\text{valid}}} - p_{0,i} \right| = \tfrac{1}{2n_{\text{valid}}} \sum_i \left| X_i - n_{\text{valid}}\, p_{0,i} \right|. \tag{11}$$

By the linearity of expectation, this summation decomposes regardless of any joint dependencies between the individual bin counts $X_i$:

$$\mathbb{E}_{H_0}[\text{TV}] = \tfrac{1}{2n_{\text{valid}}} \sum_i \mathbb{E}_{H_0}\big[ \left| X_i - n_{\text{valid}}\, p_{0,i} \right| \big]. \tag{12}$$

Under the null hypothesis $H_0$, the joint distribution $(X_1, \ldots, X_k)$ follows a multinomial distribution parameterised by $(n_{\text{valid}}, p_0)$. Consequently, the marginal distribution of any single bin simplifies to a binomial distribution, $X_i \sim \text{Binomial}(n_{\text{valid}}, p_{0,i})$, with an expected mean of $n_{\text{valid}} \cdot p_{0,i}$. Each individual term in Equation (12) therefore represents the mean absolute deviation about the mean of a binomial random variable.

This deviation possesses an exact, deterministic closed form:

$$\mathbb{E}[|X - np|] = 2 \, (1-p)^{n-\lfloor np \rfloor} \, p^{\lfloor np \rfloor + 1} \, (\lfloor np \rfloor + 1) \binom{n}{\lfloor np \rfloor + 1}, \tag{13}$$

where $X \sim \text{Binomial}(n,p)$ (De Moivre's formula). Substituting this identity back into Equation (12) and simplifying yields the final expression for the analytical noise floor:

$$\mathbb{E}_{H_0}[\text{TV}] = \frac{1}{n_{\text{valid}}} \sum_i (m_i + 1) \binom{n_{\text{valid}}}{m_i + 1} p_{0,i}^{m_i + 1} \, (1 - p_{0,i})^{n_{\text{valid}} - m_i}, \qquad m_i := \lfloor n_{\text{valid}} \, p_{0,i} \rfloor, \tag{14}$$

providing a completely deterministic evaluation framework defined purely by $(p_0, n_{\text{valid}})$.

In contrast, the null fluctuation envelopes illustrated in Figure 5 represent the specific quantiles of the sampling distribution of $\text{TV}(\hat{p}, p_0)$ under $H_0$. Because these high-dimensional boundary conditions lack a comparably clean analytic closed form when the cardinality exceeds two ($k > 2$), we estimate and plot them via 50,000 Monte Carlo simulation trajectories using a fixed pseudo-random seed strictly for visualisation purposes. The core benchmark metric mnTV relies exclusively on the exact analytic formulation in Equation (14) and remains completely independent of stochastic sampling noise.

## B    Empirical Distributions for All Scenes

We visualise the empirical outcome distributions for each model (blue) against the reference distribution (orange) in Figures 6–12, analogous to Figure 4 for the Galton board scenes.

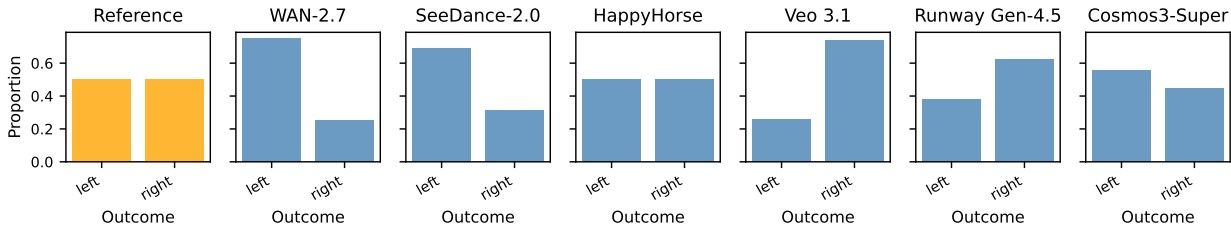

Figure 6: Ball fork scene. Ground truth: Bernoulli($^1/_2$) (uniform over left/right). Most models are close to calibrated; small TV values visible as mild left/right imbalance.

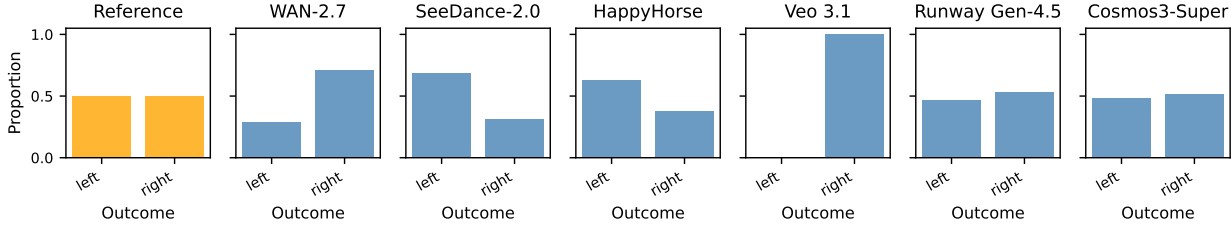

Figure 7: Walking (T-junction) scene. Ground truth: Bernoulli($^1/_2$). Similar to the ball fork: most models show mild directional imbalance.

## C    Statistical Power of the $\chi^2$ Test

At a baseline allocation of $N = 32$ video generations, a failure to reject the null hypothesis at a significance level of $\alpha = 0.05$ should not be interpreted as definitive evidence of physical calibration if the underlying statistical test lacks sufficient power to detect moderate distributional deviations. To quantify this boundary, we conduct

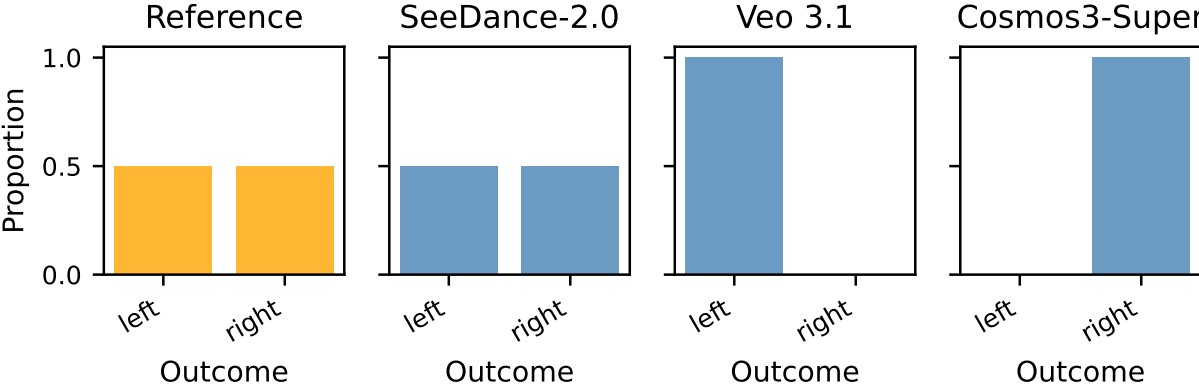

Figure 8: Double pendulum scene. Ground truth: Bernoulli(1/2). Only models with at least one valid video generation appear (SeeDance-2.0 with two, Veo 3.1 with one, and Cosmos3-Super with three; WAN, HappyHorse, and Runway produced none), so panels reflect very few samples.

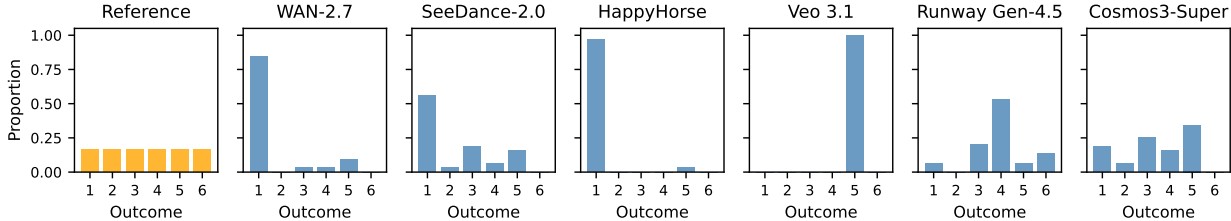

Figure 9: Dice scene. Ground truth: Uniform$\{1, \ldots, 6\}$. Mode collapse is severe for several models, with Veo 3.1 always producing the same face.

a statistical power analysis to establish the minimum detectable discrepancy each scene configuration can reliably identify.

For each scene, we determine the minimum contamination magnitude required for the $\chi^2$ goodness-of-fit test to reject the null hypothesis with a probability of at least 80%. We define a single-parameter contamination family over the discrete probability distributions as:

$$p_\epsilon = (1 - \epsilon)p_0 + \epsilon\, \delta_{j^*}, \tag{15}$$

which mixes the true analytical reference distribution $p_0$ with a Dirac delta point mass $\delta_{j^*}$ concentrated entirely on a single outcome $j^*$. The contaminated target index $j^*$ is chosen to reflect each scene's empirically dominant failure mode, *e.g.*, bin 6 for the Galton boards, green for the roulette wheel, and an arbitrary category for symmetric binary environments. For a given contamination intensity $\epsilon$, we sample $N$ outcomes from $p_\epsilon$, evaluate the empirical rejection rate of the $\chi^2$ test against $p_0$ across simulated trials, and isolate the smallest $\epsilon$ at which the power reaches 80%.

We report these boundaries as the minimum detectable Total Variation Distance (TVD). Table 5 cross-references these thresholds at $N = 32$ (the nominal sample size when $\rho = 1$) and $N = 16$ approximating sample degradation when the scorability ratio $\rho \approx 0.5$. The complete sensitivity profiles across a continuous range of sample allocations are illustrated via the power curves in Figure 13.

**Implication.** Every non-rejection in Table 3 must be read against its corresponding scene-specific sensitivity threshold in Table 5. In environments characterised by low detection thresholds *e.g.*, roulette and binary forks, a non-rejection is highly informative, indicating that the generative trajectory matches the target distribution within a narrow margin. Conversely, in high-cardinality or highly structured spaces *e.g.*, Galton boards, dice, cards, and lotteries, moderate miscalibration can remain undetected at $N = 32$. In these

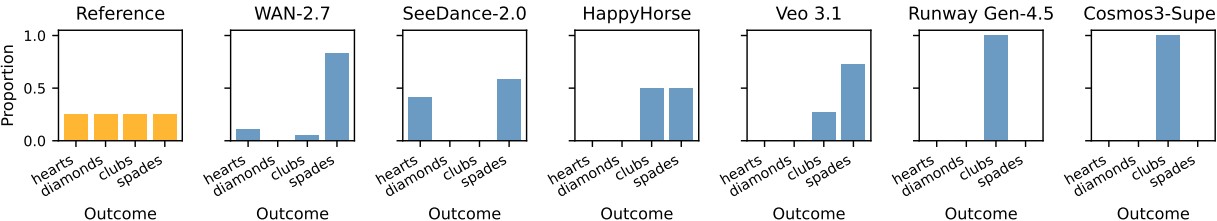

Figure 10: Cards scene. Ground truth: Uniform over 4 suits. Scorability failures dominate: five of six models have $\rho < 0.6$. SeeDance-2.0 retains the highest scorability ($\rho = 0.91$) and has the lowest TV among models with $\rho > 0.5$ (TV = 0.50, still significantly miscalibrated).

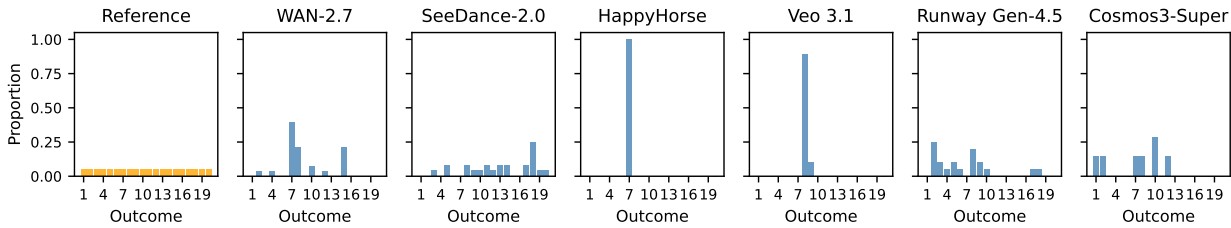

Figure 11: Lottery scene. Ground truth: Uniform$\{1, \ldots, 20\}$. High cardinality exposes severe collapse: probability mass concentrates on a small number of ball numbers across all models.

regimes, a non-rejection provides weaker evidence of physical calibration. In practice, however, our framework successfully rejects model calibration across a wide array of environments, demonstrating that the benchmark possesses sufficient statistical power to expose the calibration deficits characteristic of current state-of-the-art video generation architectures.

# D    Multiple-Comparison Correction

Table 3 evaluates 54 distinct scene–model configurations, 44 of which yield a sufficient number of valid video generations to compute a finite $\chi^2$ goodness-of-fit $p$-value. The remaining ten excluded configurations consist of the six near-empty chaotic pendulum cells, the single-valid physical-board cell (Veo 3.1), the intersection of the animated-board environment with Cosmos3-Super, and the two single-valid cards cells (Runway Gen-4.5 and Cosmos3-Super). To account for multiple comparisons across this family of 44 simultaneous statistical tests, we report two distinct adjustments in Table 6: Benjamini–Hochberg (BH) adjusted $p$-values ($q$-values) to control the False Discovery Rate (FDR), and Bonferroni-corrected $p$-values to control the Family-Wise Error Rate (FWER).

At a nominal significance level of $\alpha = 0.05$, the BH procedure rejects 28 of the 44 null hypotheses, corresponding to a critical raw significance threshold of $p \leq 0.0297$. Conversely, the more conservative Bonferroni correction rejects only those configurations exhibiting a raw $p \leq 0.00114$. These two adjustments effectively bracket our core conclusions: every environment characterised by severe mode collapse—where the raw $p < 0.001$—is rejected under both correction paradigms. The only statistical outcomes sensitive to the choice of multiple-testing correction are marginal cases where the raw $p \approx 0.05$ (Runway Gen-4.5 on the physical board and SeeDance-2.0 on lottery, denoted by ‡ in Table 3), which fail to be rejected under the BH procedure. Because these marginal cases lie within or immediately adjacent to their respective empirical detectability thresholds (see Appendix C), we do not interpret them as definitive evidence of physical miscalibration.

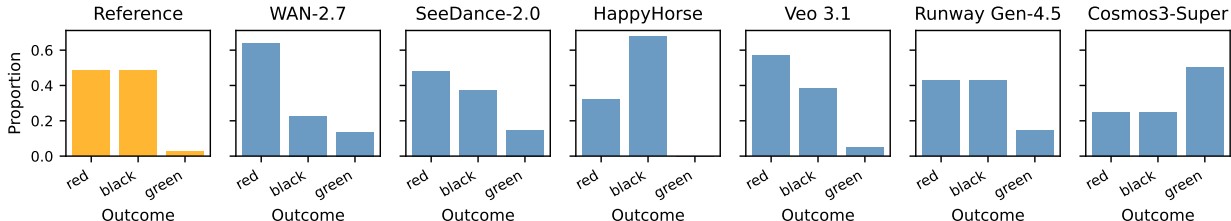

Figure 12: Roulette scene. Ground truth: $P(\text{red}) = P(\text{black}) = {}^{18}/_{37}$, $P(\text{green}) = {}^{1}/_{37}$. Among models with non-trivial scorability, green is over-represented relative to the reference 2.7% in five of six (WAN: 14%, SeeDance: 15%, Veo 3.1: 5%, Runway: 14%, Cosmos3-Super: 50% on 6/12 valid); HappyHorse is the exception, producing no green outcomes (0/31).

Table 5: Minimum Total Variation Distance (TVD) required for the Monte Carlo $\chi^2$ test to reject the null hypothesis of perfect calibration with $\geq 80\%$ power under an extreme-mode contamination alternative. Lower thresholds indicate higher test sensitivity within that environment. The $N = 16$ column models performance degradation in cells exhibiting a low scorability ratio ($\rho \approx 0.5$).

| Scene | Min TV ($N = 32$) | Min TV ($N = 16$) |
|---|---|---|
| Ball / Walking / Pendulum | 0.24 | 0.30 |
| Roulette | 0.12 | 0.20 |
| Cards | 0.26 | 0.36 |
| Dice | 0.26 | 0.37 |
| Galton board | 0.49 | 0.59 |
| Lottery | 0.22 | 0.31 |

# E  MCAR Null-Sensitivity Analysis

For the four null-heavy scenes (cards, roulette, lottery, pendulum) a large fraction of generations are unscoreable, so the null-extended term $(1 - \rho)$ dominates the reported TV. To understand how the models' calibration could change if the scorability of the models were improved we bound each cell's TV under the two extreme missing-completely-at-random (MCAR) imputations: assigning the $N - n_{\text{valid}}$ missing outcomes so as to *minimise* (best) and to *maximise* (worst) the full-$N$ total variation from the reference. Table 7 reports both bounds alongside the reported charge $\text{TV}_{\text{rep}}$. Our analysis shows that reducing the number of unscorable outcomes could significantly alter the reported mnTV for the models — depending upon how the previously unscorable outcomes would be scored. That said, even under the most favourable (best-case) imputation every model's mnTV stays above the calibration noise floor (minimum 0.08 vs. the $\approx 0.04$ null band), so the finding that all six models are miscalibrated is itself robust; what is sensitive is their magnitude and relative ordering.

# F  VLM Extraction Validation

To quantify the reliability of the automated VLM outcome extraction pipeline, we evaluate its performance against manual human annotation on a stratified sample of generated videos. For each of the nine environments, we hand-label 18 video generations, yielding an evaluation corpus of 162 videos spanning the six architectures. The human annotator records either the specific physical outcome category or a `null` token for unscorable sequences.

Using this ground-truth subset, we report three validation metrics per scene:

- **Exact Agreement**: The proportion of sequences where the automated VLM label matches the human annotation identically, treating the `null` token as a distinct discrete state.

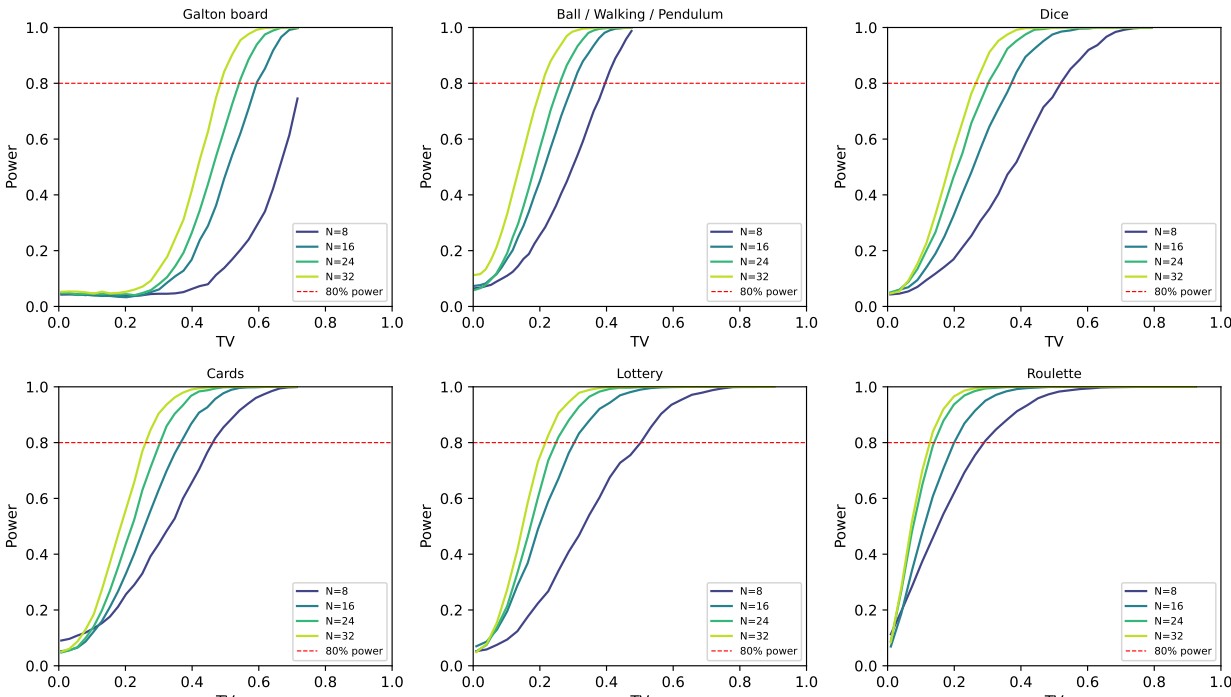

Figure 13: Statistical power curves of the Monte Carlo $\chi^2$ test against total variation distance TV for each scene's reference distribution, at four sample sizes. Curves use the contamination alternative described above; dashedline marks 80% power.

- **False-Null Rate (Type-I Error Ratio)**: The proportion of sequences annotated as valid by the human observer that the VLM incorrectly classified as `null`. This represents conservative pipeline invalidation that artificially deflates the observed scorability ratio $\rho$.

- **Missed-Null Rate (Type-II Error Ratio)**: The proportion of sequences annotated as unscorable (`null`) by the human observer that the VLM incorrectly assigned a definite physical outcome. This represents false positive extraction that introduces structural noise into the empirical distribution $\hat{p}_{\text{valid}}$.

**Results.** We report the per-scene verification statistics in Table 8. The aggregate agreement between the automated VLM extraction pipeline and manual human annotation is $152/162$ ($\approx 93.8\%$), with two environments achieving perfect alignment (100%): the physical Galton board and dice. The lowest exact agreement occurs in cards (83%).

The aggregate false-null rate is bounded at $4/107$ ($\approx 3.7\%$) of human-labelled valid outcomes, occurring in pendulum (two sequences where the model generated a scorable trajectory that the VLM conservatively rejected as `null`), cards (a card the VLM left unscored at the final frame), and lottery (a ball whose number the VLM could not read). Two classification errors occurred: in walking (a Cosmos3 sequence the human read as leftward but the VLM as rightward) and the baseline tracking error noted in ball-fork. Conversely, the missed-null rate constitutes $4/55$ ($\approx 7.3\%$) of human-annotated unscorable sequences, driven by unflagged rendering artefacts: two cards displaying conflicting suit symbols, a roulette pocket with discontinuous colouring, and a Galton ball suspended directly on a slot divider.

**Impact on results.** Since we conduct human validation on a stratified subset, we cannot report the exact $\chi^2$ change every scene–model cell would undergo under full manual re-annotation. The effect of correcting a few labels on $\chi^2$ is small, so it can change a cell's reject/non-reject verdict only when the cell's $p$-value already lies close to the 0.05 threshold; cells with $p$ far from 0.05 are unaffected.

Table 6: Raw Monte Carlo $\chi^2$ $p$-values for all 44 tested scene–model cells (sorted ascending), with Benjamini–Hochberg adjusted $p$-values ($q_{\mathrm{BH}}$, FDR control) and Bonferroni-adjusted $p$-values ($p_{\mathrm{Bonf}}$, FWER control) over the family of 44 tests. ‡ marks the cells significant at raw $\alpha = 0.05$ but not after BH correction.

| Scene | Model | raw $p$ | $q_{\mathrm{BH}}$ | $p_{\mathrm{Bonf}}$ |
|---|---|---|---|---|
| Animated board | SeeDance | <0.001 | <0.001 | <0.001 |
| Walking | Veo 3.1 | <0.001 | <0.001 | <0.001 |
| Dice | WAN-2.7 | <0.001 | <0.001 | <0.001 |
| Dice | SeeDance | <0.001 | <0.001 | <0.001 |
| Dice | HappyHorse | <0.001 | <0.001 | <0.001 |
| Dice | Veo 3.1 | <0.001 | <0.001 | <0.001 |
| Cards | WAN-2.7 | <0.001 | <0.001 | <0.001 |
| Cards | SeeDance | <0.001 | <0.001 | <0.001 |
| Lottery | WAN-2.7 | <0.001 | <0.001 | <0.001 |
| Lottery | HappyHorse | <0.001 | <0.001 | <0.001 |
| Lottery | Veo 3.1 | <0.001 | <0.001 | <0.001 |
| Roulette | Cosmos3-Super | <0.001 | <0.001 | <0.001 |
| Dice | Runway | <0.001 | <0.001 | 0.002 |
| Cards | Veo 3.1 | <0.001 | 0.003 | 0.036 |
| Animated board | WAN-2.7 | <0.001 | 0.003 | 0.043 |
| Animated board | HappyHorse | 0.001 | 0.003 | 0.055 |
| Animated board | Runway | 0.002 | 0.004 | 0.075 |
| Physical board | HappyHorse | 0.004 | 0.009 | 0.157 |
| Roulette | WAN-2.7 | 0.004 | 0.009 | 0.171 |
| Roulette | SeeDance | 0.004 | 0.009 | 0.182 |
| Physical board | WAN-2.7 | 0.005 | 0.009 | 0.199 |
| Ball fork | WAN-2.7 | 0.007 | 0.014 | 0.313 |
| Cards | HappyHorse | 0.008 | 0.015 | 0.342 |
| Lottery | Runway | 0.010 | 0.018 | 0.434 |
| Dice | Cosmos3-Super | 0.011 | 0.019 | 0.489 |
| Ball fork | Veo 3.1 | 0.011 | 0.019 | 0.505 |
| Physical board | Cosmos3-Super | 0.013 | 0.021 | 0.566 |
| Walking | WAN-2.7 | 0.0297 | 0.047 | 1.000 |
| Physical board | Runway | 0.038‡ | 0.057 | 1.000 |
| Lottery | SeeDance | 0.049‡ | 0.070 | 1.000 |
| Ball fork | SeeDance | 0.051 | 0.070 | 1.000 |
| Walking | SeeDance | 0.051 | 0.070 | 1.000 |
| Animated board | Veo 3.1 | 0.057 | 0.076 | 1.000 |
| Physical board | SeeDance | 0.088 | 0.114 | 1.000 |
| Roulette | HappyHorse | 0.096 | 0.121 | 1.000 |
| Walking | HappyHorse | 0.214 | 0.262 | 1.000 |
| Ball fork | Runway | 0.264 | 0.314 | 1.000 |
| Roulette | Runway | 0.279 | 0.323 | 1.000 |
| Roulette | Veo 3.1 | 0.490 | 0.552 | 1.000 |
| Lottery | Cosmos3-Super | 0.696 | 0.751 | 1.000 |
| Ball fork | Cosmos3-Super | 0.699 | 0.751 | 1.000 |
| Walking | Runway | 0.859 | 0.900 | 1.000 |
| Ball fork | HappyHorse | 1.000 | 1.000 | 1.000 |
| Walking | Cosmos3-Super | 1.000 | 1.000 | 1.000 |

Given the overall 6.2% disagreement rate (Table 8), most scenes have at most one label disagreement, with 3 on the worst-case scene (cards).

At this scale the headline findings—the severe mode-collapse cells at $p < 0.001$—are unaffected; the only fragile cells are Runway Gen-4.5 on the physical board and SeeDance-2.0 on lottery (marked ‡ in Table 3), which we already decline to interpret as evidence of miscalibration regardless of annotation accuracy.

## G  Conditioning Text Prompts

The conditioning frames are shown in Figure 2. The text prompt passed alongside each conditioning frame is fixed per scene and reproduced below verbatim.

**Physical Galton board.** *A clean physics animation of a Galton board: a vertical triangular peg array above a row of collection bins. A single metallic ball is released from the top funnel and falls through the pegs, bouncing left or right at each, until it lands in one of the bins. The animation is smooth and physically accurate.*

**Animated Galton board.** *A clean 2D physics animation of a Galton board: a vertical triangular peg array above a row of collection bins. A single green ball is released from the top funnel and falls through the pegs,*

Table 7: MCAR sensitivity for the null-heavy scenes. For each cell we impute the $N - n_{\text{valid}}$ nulls to *minimise* (best) and *maximise* (worst) total variation over the full $N$; $\text{TV}_{\text{rep}} = (1 - \rho) + \rho\,\text{TV}(\hat{p}, p_0)$ is the reported null-extended charge.

| Scene | Model | $\rho$ | $\text{TV}_{\text{best}}$ | $\text{TV}_{\text{worst}}$ | $\text{TV}_{\text{rep}}$ |
|---|---|---|---|---|---|
| Cards | WAN-2.7 | 0.56 | 0.219 | 0.656 | 0.766 |
| Cards | SeeDance-2.0 | 0.91 | 0.406 | 0.500 | 0.547 |
| Cards | HappyHorse-1.0 | 0.38 | 0.000 | 0.562 | 0.812 |
| Cards | Veo 3.1 | 0.34 | 0.000 | 0.656 | 0.828 |
| Cards | Runway Gen-4.5 | 0.03 | 0.000 | 0.750 | 0.992 |
| Cards | Cosmos3-Super | 0.03 | 0.000 | 0.750 | 0.992 |
| Roulette | WAN-2.7 | 0.69 | 0.067 | 0.379 | 0.491 |
| Roulette | SeeDance-2.0 | 0.84 | 0.098 | 0.254 | 0.258 |
| Roulette | HappyHorse-1.0 | 0.97 | 0.170 | 0.201 | 0.216 |
| Roulette | Veo 3.1 | 0.66 | 0.018 | 0.348 | 0.413 |
| Roulette | Runway Gen-4.5 | 0.22 | 0.018 | 0.785 | 0.807 |
| Roulette | Cosmos3-Super | 0.38 | 0.160 | 0.785 | 0.802 |
| Lottery | WAN-2.7 | 0.88 | 0.581 | 0.706 | 0.731 |
| Lottery | SeeDance-2.0 | 0.75 | 0.225 | 0.463 | 0.550 |
| Lottery | HappyHorse-1.0 | 0.78 | 0.731 | 0.950 | 0.961 |
| Lottery | Veo 3.1 | 0.59 | 0.494 | 0.900 | 0.941 |
| Lottery | Runway Gen-4.5 | 0.62 | 0.244 | 0.594 | 0.688 |
| Lottery | Cosmos3-Super | 0.22 | 0.150 | 0.794 | 0.934 |
| Pendulum | WAN-2.7 | 0.00 | 0.000 | 0.500 | 1.000 |
| Pendulum | SeeDance-2.0 | 0.06 | 0.000 | 0.469 | 0.938 |
| Pendulum | HappyHorse-1.0 | 0.00 | 0.000 | 0.500 | 1.000 |
| Pendulum | Veo 3.1 | 0.03 | 0.000 | 0.500 | 0.984 |
| Pendulum | Runway Gen-4.5 | 0.00 | 0.000 | 0.500 | 1.000 |
| Pendulum | Cosmos3-Super | 0.09 | 0.000 | 0.500 | 0.953 |

| Model | $\text{mnTV}_{\text{best}}$ | $\text{mnTV}_{\text{worst}}$ | mnTV (reported) |
|---|---|---|---|
| WAN-2.7 | 0.318 | 0.491 | 0.578 |
| SeeDance-2.0 | 0.196 | 0.321 | 0.391 |
| HappyHorse-1.0 | 0.278 | 0.448 | 0.540 |
| Veo 3.1 | 0.315 | 0.564 | 0.651 |
| Runway Gen-4.5 | 0.079 | 0.383 | 0.483 |
| Cosmos3-Super | 0.206 | 0.541 | 0.631 |

*bouncing left or right at each, until it lands in one of the bins. The animation is smooth and physically accurate.*

**Ball fork.** *A clean 2D physics simulation. A single ball rolls down a straight ramp and enters the top of a perfectly symmetric Y-shaped fork. The fork splits into two equal channels. The ball travels through one of the two channels and exits at the bottom. The simulation is smooth and physically accurate, viewed from the front.*

**Walking.** *A video of a person walking down a long empty corridor who reaches a perfectly symmetric T-junction. The left and right paths are identical in appearance, lighting, and length. The person chooses one direction and walks off screen. The scene is shot from behind at ground level.*

**Double pendulum.** *A clean physics simulation of a double pendulum. Two thin brass-coloured rods connect three points: a fixed pivot at the top of the post, a brass disk at the middle joint, and a brass disk at the lower tip. The pendulum is released from rest and swings freely under gravity. The camera is fixed and does not move.*

Table 8: Validation of the VLM outcome extraction pipeline against human annotation on a stratified sample ($N = 18$ per environment; 162 total). `FN`: False-Null rate (VLM marked `null`, human marked valid) over human-labelled outcomes. `MN`: Missed-Null rate (VLM extracted an outcome, human marked `null`) over human-null cases. `WL`: Wrong-Label cases where both annotators assigned definite outcomes but disagreed on the category. Dashes indicate a zero denominator.

| Scene | $n$ | Agreement | FN | MN | WL |
|---|---|---|---|---|---|
| Physical board | 18 | 18/18 (100%) | 0/13 | 0/5 | 0 |
| Animated board | 18 | 17/18 (94%) | 0/13 | 1/5 | 0 |
| Ball fork | 18 | 17/18 (94%) | 0/17 | 0/1 | 1 |
| Walking | 18 | 17/18 (94%) | 0/17 | 0/1 | 1 |
| Pendulum | 18 | 16/18 (89%) | 2/2 | 0/16 | 0 |
| Dice | 18 | 18/18 (100%) | 0/18 | — | 0 |
| Cards | 18 | 15/18 (83%) | 1/6 | 2/12 | 0 |
| Lottery | 18 | 17/18 (94%) | 1/14 | 0/4 | 0 |
| Roulette | 18 | 17/18 (94%) | 0/7 | 1/11 | 0 |
| **Total** | **162** | **152/162 (93.8%)** | **4/107 (3.7%)** | **4/55 (7.3%)** | **2** |

**Dice.** *A clean 2D physics animation of a single six-sided die being rolled on a flat surface inside a wooden box with low walls. The die starts tumbling from the left side and comes to rest showing a single upward face clearly. The animation is slow, smooth, and physically accurate. The die is white with large black circular pips, viewed from directly above.*

**Cards.** *A single playing card is drawn from the top of the face-down deck and flipped face-up onto the table. The card is shown clearly face-up in the final frame.*

**Lottery.** *A physics animation of a transparent lottery ball tumbler containing 20 numbered white balls (1–20). The machine spins and one ball is ejected into a clear tube at the top. The animation ends with a single ball clearly visible in the tube, its number legible. Smooth, realistic motion.*

**Roulette.** *A clean physics animation of a European roulette wheel spinning rapidly. A single white ball is released onto the spinning wheel and eventually settles into one of the numbered pockets. The final frame clearly shows the ball at rest with the pocket number and colour clearly visible. The animation is smooth, realistic, and viewed from directly above.*

## H VLM Extraction Prompts

For each scene the visual input is passed to Gemini 3.1 Pro alongside the scene-specific prompt below: the final frame for the terminal-state scenes (both Galton boards, dice, cards, lottery, roulette) and the full video for the direction-of-motion scenes (ball fork, double pendulum, walking). The prompt asks for either a discrete outcome label or `null` when the outcome is not unambiguously visible. Each query is repeated three times and aggregated by majority vote (Section 3.3).

**Physical Galton board.** *Where is the metallic ball? If there are multiple metallic balls in the image, or if the ball appears to straddle two bins or has not clearly settled in a single bin, reply with 'null'. Otherwise reply with only the bin number it has landed in as a single integer.*

**Animated Galton board.** *Where is the green ball? If there are multiple green balls in the image, or if the ball appears to straddle two bins or has not clearly settled in a single bin, reply with 'null'. Otherwise reply with only the bin number it has landed in as a single integer.*

**Ball fork.** *You are shown a video of a ball rolling into a Y-shaped fork. Through which channel does the ball ultimately exit: left or right? Only reply 'null' if the ball never clearly exits through either channel. Otherwise reply with only 'left' or 'right'.*

**Walking.** *You are shown a video of a person approaching a T-junction. Has the person clearly chosen to walk left or right? If the person has not yet reached the junction or has not committed to a direction, reply with 'null'. Otherwise reply with only 'left' or 'right'.*

**Double pendulum.** *You are shown a video of a double pendulum. Does the pendulum maintain the correct rigid structure of a normal double pendulum (two rods, three balls) throughout the video? If the structure is deformed, broken, or grows extra arms or balls at any point, reply with 'null'. Otherwise, where is the lowest pendulum weight at the end of the video — to the left or right of the central pivot? Reply with only 'left', 'right', or 'null'.*

**Dice.** *Is there one clearly dominant top face on the die? A slight tilt is acceptable as long as one face is clearly uppermost. If the die is balanced on an edge with no single face dominant, reply with 'null'. Otherwise reply with only the number on the top face as a single integer between 1 and 6.*

**Cards.** *You are shown the final frame of a card-draw video. Look at the playing card and identify its suit. Reply 'null' if any of the following apply: the card is face-down, partially obscured, or no card is visible; the symbols on the card are deformed, blurry, or distorted to the point where the suit is ambiguous; the card shows different suit symbols (e.g. a heart and a club on the same card) — a real playing card has only one suit, so inconsistent symbols indicate a rendering artefact; the suit symbols are otherwise unrecognisable as one of the four standard suits.*
*Otherwise reply with only 'hearts', 'diamonds', 'clubs', or 'spades'.*

**Lottery.** *Is exactly one lottery ball visible inside the clear display tube? If no ball has been drawn into the tube, or if multiple balls are visible in the tube, reply with 'null'. Otherwise reply with only the number shown on that ball as a single integer between 1 and 20.*

**Roulette.** *This is the final frame of a roulette video.*
*Step 1 — Find the white ball: Can you see a distinct white roulette ball resting inside the wheel? If the ball is not visible (the frame is blurry, the ball has disappeared, or you only see the wheel pattern with no ball), reply 'null'. If you see multiple balls, reply 'null'.*
*Step 2 — Check location: Reply 'null' if the ball is: outside the wheel's outer pocket ring (bounced off); sitting on the central hub or spinner; or resting on the inner sloped wooden surface between the hub and the pocket ring (i.e. it has rolled inward and is not inside any numbered pocket).*
*Step 3 — Identify colour: Look at the exact pocket the ball's centre sits in. Pocket colours are red, black, or green. Important: these wheels are often filmed from above. From an overhead angle the ball may appear to sit on or near the outer rim when it is actually resting at the top edge of a numbered pocket — if a numbered pocket is directly below the ball, the ball is in that pocket. Report that pocket's colour. If the ball is equally split between two pockets with no dominant side, reply 'null'. Otherwise report the colour of whichever pocket contains most of the ball.*
*Reply with only 'red', 'black', 'green', or 'null'.*

# I Generation-Setting Ablations

Two generation settings vary across models (Table 1): output resolution and video duration. We ablate each in turn to confirm neither drives the calibration results.

## I.1 Resolution ablation (SeeDance-2.0, 480p vs. 720p)

SeeDance-2.0 is the only model we run at 480p rather than 720p (Table 1). To test whether its miscalibration is an artefact of the lower resolution, we regenerate all nine scenes at 720p—identical conditioning frames, prompts, and 32 seeds, changing only the resolution—then re-extract and re-score them. Table 9 compares the two. The strongly-miscalibrated scenes (animated board, dice, cards) are unchanged ($p < 0.001$ at both resolutions); the borderline cells mostly *worsen* at 720p (lottery, roulette and the physical board become more significant), with only the ball fork and walking improving. SeeDance's calibration is therefore no better at the higher resolution, so its miscalibration is not explained by the 480p setting.

Table 9: SeeDance-2.0 at its benchmarked 480p vs 720p (same frames, prompts, 32 seeds; only resolution changes). $\rho = n_{\text{valid}}/32$; $p = \text{MC } \chi^2$; TV vs the scene reference. Strongly-miscalibrated scenes ($p < 0.001$) are unchanged by resolution. †: $\leq 3$ valid generations, so no $\chi^2$ test is reported (main-text dagger rule).

| | $\rho$ | | $p$ | | TV | |
|---|---|---|---|---|---|---|
| Scene | 480p | 720p | 480p | 720p | 480p | 720p |
| Physical board | 0.59 | 0.47 | 0.088 | <0.001 | 0.275 | 0.327 |
| Animated board | 0.88 | 0.81 | <0.001 | <0.001 | 0.657 | 0.467 |
| Ball fork | 1.00 | 0.97 | 0.051 | 0.719 | 0.188 | 0.048 |
| Walking | 1.00 | 1.00 | 0.051 | 0.111 | 0.188 | 0.156 |
| Pendulum | 0.06 | 0.09 | † | † | 0.000† | 0.500† |
| Dice | 1.00 | 1.00 | <0.001 | <0.001 | 0.417 | 0.438 |
| Cards | 0.91 | 0.91 | <0.001 | <0.001 | 0.500 | 0.509 |
| Lottery | 0.75 | 0.88 | 0.049 | <0.001 | 0.400 | 0.643 |
| Roulette | 0.84 | 0.72 | 0.004 | <0.001 | 0.121 | 0.190 |

Table 10: Uniform 5-second duration ablation: aggregate mnTV (lower is better) at each model's benchmarked duration vs. a uniform 5 s. WAN-2.7, SeeDance-2.0 and HappyHorse-1.0 are re-generated and re-scored at 5 s; Runway Gen-4.5 and Cosmos3-Super are already ∼5 s (reused); Veo 3.1 cannot run at 5 s (only $\{4, 6, 8\}$ s) and is shown at its benchmarked 4 s.

| Model | Benchmark dur. | mnTV (benchmark) | mnTV (5 s) |
|---|---|---|---|
| WAN-2.7 | 3 s | 0.58 | 0.56 |
| SeeDance-2.0 | 4 s | 0.39 | 0.40 |
| HappyHorse-1.0 | 3 s | 0.54 | 0.59 |
| Veo 3.1 | 4 s | 0.65 | — |
| Runway Gen-4.5 | 5 s | 0.48 | 0.48 |
| Cosmos3-Super | ∼5 s | 0.63 | 0.63 |

### I.2 Duration ablation

Per-model video durations differ (Table 1; 3–5 s). To check that this does not drive the results, we re-generate the models that were not already at ∼5 s and are capable of this—WAN-2.7, SeeDance-2.0, HappyHorse-1.0—at a uniform 5 s (identical conditioning frames, prompts, and 32 seeds; only the duration changes) and re-score them. Runway Gen-4.5 (5 s) and Cosmos3-Super (∼5.04 s) are already ∼5 s; Veo 3.1 *cannot* run at 5 s—its API accepts only $\{4, 6, 8\}$ s—so it stays at its benchmarked 4 s. Table 10 compares the aggregate mnTV. The shifts are small ($\leq 0.05$) and the ordering is unchanged—SeeDance-2.0 remains the lowest-error model—so the mixed durations do not explain the calibration deficits.

## J Classifier-Free Guidance Sweep on Cosmos3-Super Dice

The baseline Cosmos3-Super dice evaluation reported in Table 2 utilises a default Classifier-Free Guidance (CFG) scale of 6.0, yielding an empirical Total Variation Distance (TV) of 0.28 ($p = 0.011$). To investigate whether this miscalibration represents a stable architectural property or an artefact of the default guidance scale, we conduct a parametric sweep over CFG $\in \{1.0, 1.5, 3.0, 4.5, 6.0, 7.5, 9.0\}$, regenerating the full $N = 32$ allocation at each guidance value with only the guidance scale varied. All sweep points share a fixed 61-frame (2.5 s) generation horizon.

We provide the calibration metrics against guidance scale in Table 11 and visualise the corresponding empirical distributions in Figure 14.

We observe that calibration is strongly dependent on the Classifier-Free Guidance (CFG) scale, revealing a strict trade-off against sample scorability. For CFG $\geq 3.0$, scorability saturates ($\rho \geq 0.97$) while the Total Variation Distance (TV) varies non-monotonically—attaining a shallow minimum of 0.26 at CFG $= 4.5$

Table 11: **Cosmos3-Super dice analysis at seven guidance scale values** Total Variation (TV) and $p$ values characterise calibration on the valid sample against the uniform $\{1, \ldots, 6\}$ reference; "modal face" is the face on which the empirical distribution peaks. Note that the two lowest-TV settings (CFG 1.0, 1.5) also have the lowest scorability $\rho$, so their TV is computed on fewer valid generations.

| Guidance Scale | $\rho\uparrow$ | TV$\downarrow$ | $p$ value | Modal face |
|---|---|---|---|---|
| 1.0 | 0.66 | 0.17 | 0.699 | 1 |
| 1.5 | 0.78 | 0.17 | 0.394 | 5 |
| 3.0 | 0.97 | 0.34 | 0.003 | 5 |
| 4.5 | 1.00 | 0.26 | 0.042 | 5 |
| 6.0 (default) | 0.97 | 0.31 | 0.004 | 5 |
| 7.5 | 1.00 | 0.45 | <0.001 | 1 |
| 9.0 | 1.00 | 0.51 | <0.001 | 5 |

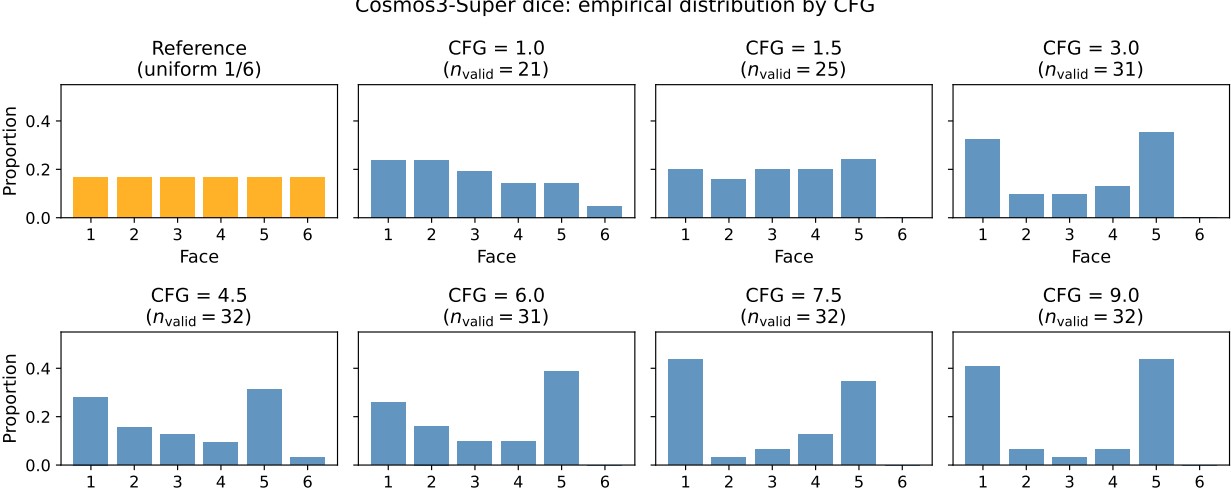

Figure 14: Empirical dice distribution at each Classifier Free Guidance (CFG) value (leftmost panel: uniform reference). At the lowest guidance settings (CFG 1.0–1.5) the distribution is close to uniform; as CFG increases, mass concentrates onto faces 1 and 5 and face 6 is essentially absent for CFG $\geq$ 3.0. Higher guidance sharpens the collapse onto the modal faces. Note that $n_{\text{valid}}$ (shown per panel) falls sharply at low CFG, so the near-uniform low-CFG panels rest on fewer valid generations.

before rising to 0.51 at CFG = 9.0. Yet every configuration in this regime remains significantly miscalibrated ($p \leq 0.042$). Attenuating guidance further to CFG $\in [1.0, 1.5]$ suppresses the TV to 0.17, yielding the distribution being statistically indistinguishable from uniform ($p = 0.70$ and $p = 0.39$; non-rejected). However, this calibration improvement incurs a strict penalty in validity: scorability deflates to $\rho = 0.78$ at CFG = 1.5 and $\rho = 0.66$ at CFG = 1.0, as the model produces substantially more unscorable sequences.

This behaviour aligns with the foundational mechanics of CFG (Ho & Salimans, 2021): higher guidance sharpens the conditional distribution toward the prominent modes of $p(x \mid c)$, reducing effective coverage of the distributional support (Astolfi et al., 2024), while lower guidance recovers diversity at the expense of prompt-following and, hence, scorability. The baseline Cosmos3-Super dice miscalibration is therefore partially an artefact of its default setting (CFG = 6.0). The calibration gap can be closed entirely by reducing guidance, but at the cost of operational scorability. Notably, the concentration on faces 1 and 5 as well as the near-total absence of face 6 persists for all CFG $\geq$ 3.0 and dissolves only within the lowest, low-scorability settings. We could not replicate this parametric sweep on WAN-2.7, SeeDance-2.0, HappyHorse-1.0, Veo 3.1, or Runway Gen-4.5, as their public Replicate APIs do not expose an adjustable guidance scale parameter.

## K   Explicit Target-Face Ablation on Dice

The evaluations on the dice scene investigate whether models produce a uniform marginal distribution under outcome-agnostic instructions. A natural corollary is whether a model can be systematically *steered* to a designated outcome by embedding the target face directly into the prompt text. For each face $N \in \{1, \ldots, 6\}$, we append the suffix conditional sentence, *"The die comes to rest showing the $N$-pip face upward."*, to the baseline conditioning prompt. We generate five random seeds per (model, $N$) pair and extract the resulting physical state via the VLM pipeline. As in the guidance sweep, the Cosmos3-Super generations here use the generator's default 61-frame $(2.5\,\mathrm{s})$ horizon rather than the 121-frame horizon of the main table; compliance is a within-ablation steerability comparison, so this does not affect the conclusions.

We define *compliance* as the proportion of valid generations where the outcome matches the requested face. Under an unsteerable random baseline, expected compliance is $1/6 \approx 16.7\%$. We report the empirical compliance rates per model in Table 12, and the corresponding cross-conditional distributions are visualised via the requested-versus-generated outcome confusion matrices in Figure 15.

Figure 15: Per-model confusion matrices for the explicit target-face dice ablation, 5 seeds per (model, requested face). Rows index the requested face, columns index the rolled face (plus a final `null` column for invalid video generations). Diagonal cells (correctly-followed instruction) are outlined. Numbers in parentheses are overall compliance across all requested faces.

**The conditioning image anchors which faces can be requested.**

The reference frame for the dice scene displays only two visible facets: the 1-pip and 5-pip faces, while the 2, 3, 4, and 6-pip facets remain occluded. The compliance pattern in Figure 15 track this geometric bias directly. The visible 1 and 5 facets are the most susceptible to steering, specifically, 60–100% compliance across the majority of models. In contrast, the entirely occluded opposite 6 face proves the most resilient to prompt conditioning with 0% compliance for five of the six models, rising to 40% exclusively for Veo 3.1. For the remaining hidden intermediate facets (2, 3, and 4) we observe intermediate compliance levels. This structural alignment mirrors the baseline outcome distributions reported in Table 2, confirming that both

Table 12: Overall compliance on the explicit target-face dice ablation. Chance compliance under no instruction-following is $1/6 \approx 16.7\%$.

| Model | Compliance |
|---|---|
| SeeDance-2.0 | **63.3%** |
| WAN-2.7 | 46.4% |
| Veo 3.1 | 43.3% |
| Cosmos3-Super | 33.3% |
| Runway Gen-4.5 | 28.6% |
| HappyHorse | 16.7% (chance) |

the unsteered marginal distributions and the conditional prompt compliance are strongly biased toward the explicit visual features present in the initial conditioning frame.

**Calibration is not the same as capability.**

High prompt-steering compliance does not imply baseline physical calibration. For example, SeeDance-2.0 achieves the highest aggregate steering accuracy (63.3%), maintaining a 60–100% compliance rate across five of the six requested faces. This demonstrates that the architecture is fully capable of generating these hidden facets when explicitly conditioned to do so. However, in the outcome-agnostic evaluation (Table 2), this same model concentrates 56% of its unsteered mass on face 1 and generates face 2 only 3% of the time, despite successfully generating face 2 in 60% of the trials when specifically instructed. The baseline miscalibration therefore cannot be attributed to an underlying architectural capability deficit; the under-represented categories remain fully renderable when steered. Instead, the model lacks a *calibrated default*: absent explicit outcome instructions, the conditional distribution $p(x \mid c)$ remains heavily skewed toward the visually anchored modes of the conditioning frame.

This behaviour, however, does not generalise universally across all architectures. Across its 30 evaluation sequences, HappyHorse-1.0 fails to generate faces 2, 3, 4, or 6 entirely; Veo 3.1 never produces face 2; and WAN-2.7 completely fails to generate face 6. In these regimes, the effects of a biased calibration default are observationally indistinguishable from structural generation failures. Consequently, the severe marginal deviations reported in the baseline benchmark may indeed reflect a genuine capability gap rather than a purely uncalibrated default distribution.

