# OpenReview forum: "CaliBench: Are the Stochastic Dynamics of Video World Models Physically Calibrated?"
_TMLR — Under review for TMLR_

### Review · Reviewer_ZJ2B · 2026-07-02

**Summary Of Contributions:**

This manuscript constructs a challenging quantitative benchmark designed to probe whether state-of-the-art video generation models truly understand the objective laws of the physical world. Moving away from traditional appearance-based metrics (e.g., FVD), this work introduces a suite of evaluation scenarios rigorously constrained by classical mechanics and probability theory (e.g., Galton board, pendulum, roulette, dice face, and lottery). This paper systematically compares leading open-source and closed-source video foundation models (e.g., Cosmos, Runway, Veo, Wan). It provides a comprehensive analysis of the performance boundaries and inherent limitations of these models in long-term physical causal reasoning.

**Audience:**

Yes

**Audience Explanation:**

1.The paper explores a massive variety of physical systems and probabilistic scenarios, such as the Galton board, pendulum dynamics, roulette, and rolling dice. Researchers studying physics-informed neural networks (PINNs), dynamical systems, or the predictability of chaotic physical processes would find the findings highly relevant to their own modeling efforts.
2. For TMLR readers specializing in generative models (such as diffusion models or autoregressive transformers), understanding how a model accurately handles specific conditioning inputs to simulate complex multi-service platform behaviors is a core area of interest.

**Broader Impact Concerns:**

There may be no ethical implications.

**Claims And Evidence:**

Yes

**Claims Explanation:**

1.The supplementary materials contain analytical data for a massive variety of physical systems and scenarios, such as the Galton board, pendulum, roulette, cards, ball movement. The data is meticulously categorized by different experiment codenames or platforms.
2. This paper did not rely solely on automated machine outputs or algorithmic simulations; they integrated "human labels" as a ground truth or evaluation benchmark. In studies involving images, videos, or behavioral analysis, cross-referencing algorithmic outcomes with actual human perception or annotations forcefully validates the accuracy of the claims.

**Requested Changes:**

1. Although some excellent environments are adopted for testing classical mechanics, these tasks are highly synthetic and visually simplistic. It is better to demonstrate whether a model’s performance on these idealized "physical toys" directly correlates with its capability to handle complex, open-world physical dynamics in realistic video generation.
2. The evaluation includes several proprietary commercial models (e.g., Veo, Runway). It is difficult to theoretically rule out the possibility that these models have already "memorized" vast amounts of similar physical simulations during pre-training. A robust analysis addressing potential "data contamination" or data leakage is missing.
3. It is better to provide a deeper architectural analysis (e.g., diffusion-based vs. transformer-based architectures) regarding why specific physical causal chains break down—such as whether the failure stems from spatial attention collapse or inadequate temporal scaling in the latent space.

---

### Review · Reviewer_PCsN · 2026-07-06

**Summary Of Contributions:**

CaliBench evaluates the following question: sample an image-to-video model repeatedly from the same conditioning frame and prompt, changing only the seed, and check whether the outcome distribution matches the known distribution of the underlying random physical process. The nine scenes are set up so that outcomes are discrete and the reference distribution has a closed form: two Galton boards (Binomial(10, 1/2)), three binary scenes (ball fork, T-junction walker, double pendulum, all Bernoulli(1/2)), three uniform scenes (dice, card suits, lottery) and European roulette colour (red and black 18/37 each, green 1/37). Six models generate 32 videos per scene. Outcomes are read out by Gemini 3.1 Pro with three-query majority voting, compared against human labels on 162 videos with 93.2% agreement. Each cell gets two numbers: scorability ρ, the fraction of valid videos, and calibration, the total variation distance TV between the valid-sample distribution and the reference. Significance uses a Monte Carlo exact chi-square test with BH and Bonferroni corrections, together with a power analysis and an aggregate score mnTV that subtracts a closed-form finite-sample noise floor. Main results: 28 of 45 testable cells remain significantly off the reference after correction, most models concentrate probability mass on few outcomes in several scenes, with Veo 3.1 producing the same die face in all 32 videos; no model leads in all scenes; lowering CFG improves calibration and reduces scorability; a steering experiment shows rare outcomes can be generated under explicit instruction, while the default distribution leans toward the faces visible in the conditioning image.

**Audience:**

Yes

**Audience Explanation:**

Evaluation of video world models is an active area, and to my knowledge no prior work tests distribution-level calibration against closed-form references in a systematic way, so the paper complements per-sample suites (VBench, PhysicsIQ and others) and feature-space metrics (FVD, JEDi). Several results are relevant to model development and evaluation design: single-sample realism can coexist with distributional collapse, which existing checks do not cover; CFG involves a trade-off between calibration and scorability; and models can be steered to rare outcomes while the default distribution remains skewed. Frames, prompts, labels, per-rollout data and code are released, so the setup can be reused. For readers considering world models for safety testing in driving or robotics, the observation that overly deterministic rollouts mask tail risk is relevant.

**Broader Impact Concerns:**

None. The motivation relates to safety evaluation, and the gambling scenes are standard probability examples, so no statement is needed.

**Claims And Evidence:**

Yes

**Claims Explanation:**

I recomputed the results from the released per-rollout data (analysis/*_bins.json) with my own code, not the authors' scripts: all 54 cells of Table 1 match on ρ and TV, the six mnTV values, the per-cell scores in Table 3, the CFG sweep in Appendix H and the steering table in Appendix I also match, and the individual results are consistent with the data (Veo 3.1 dice 32 of 32 on one face, pendulum valid counts 1, 0, 1, 0, 0 and 8, roulette green rates 14%, 15%, 0%, 5%, 14% and 50% against a 2.7% reference). On the statistical side, the paper uses an exact test where the asymptotic one does not apply, reports effect size separately from significance, applies two corrections and includes a power analysis; I found no methodological errors there. On this basis the central claims are consistent with the released data. The yes carries four caveats. First, at N=32 the test only detects large deviations; the minimum detectable TV at 80% power is 0.49 on the Galton boards, the abstract and results section do not mention this limit, and non-rejected cells cannot be read as calibrated. Second, the pendulum reference Bernoulli(1/2) is a design choice rather than a derived result, the paper states the chaotic regime was not verified, the released frame has an asymmetric starting configuration, and with five of six models at ρ of 0.03 or less the scene functions as a structural integrity test that contributes close to 1 to mnTV. Third, each scene uses a single conditioning frame while Appendix I shows the frame affects the outcome distribution: the dice image shows only faces 1 and 5, and the hidden face 6 has zero steering compliance on five of six models, so scene-level conclusions carry no uncertainty estimate over the stimulus. Fourth, the released code shows differing generation settings across models (SeeDance at 480p, durations from 3 to 5 seconds, prompt expansion on for WAN, no Cosmos3-Super pipeline in the release), which affects the interpretation of cross-model comparisons and the mnTV ranking. I also spot-checked five p-values from Table 2: four agree within sampling error, and one reflects an implementation issue in stats_utils.py where the Monte Carlo p-value compares floats and excludes ties, so the lottery and SeeDance cell reports 0.034 where the tie-inclusive value is 0.049; results at p below 0.001 are unaffected, and borderline cells need recomputation with the add-one correction of Hope (1968).

**Requested Changes:**

Five critical items, suggested as conditions for acceptance: first, state in the abstract and at the main claims that N=32 only reliably detects large deviations (citing the per-scene floors in Table 4) and that non-rejection does not imply calibration; second, provide a quantitative argument for the pendulum 50/50 reference (for example a perturbation simulation of the released frame) or classify it as a structural test and report mnTV with and without it; third, list the per-model duration, resolution and prompt expansion settings in the main text, quantify the effect on at least one scene (for example SeeDance at 480p versus 720p), release the Cosmos3-Super generation config, and adjust the cross-model ranking statements accordingly; fourth, correct the tie handling in the Monte Carlo p-values using the add-one estimator of Hope (1968) with tolerNone. The motivation relates to safety evaluation, and the gambling scenes are standard probability examples, so no statement is needed.
ance-based counting and recompute Tables 2 and 5, which leaves results at p below 0.001 unchanged while borderline cells and annotations need updating; fifth, run an MCAR sensitivity analysis for the cells with many nulls (cards, roulette, lottery, pendulum), assigning nulls in the best and worst case and reporting the resulting TV range. Recommended, not blocking: evaluate two or three conditioning frames per scene; release raw VLM responses and vote logs, add a second annotator and report agreement, pin the extractor version; fix the code issues (API exceptions are recorded as nulls and lower ρ, the votes default of 1 differs from the 3 used in the paper, parsing takes the first matching substring or integer and fails on longer replies, constants are duplicated across five scripts, the supplement contains system temporary files); clarify that held static across generations refers to a single cell, and correct the typos To evalute (Section 3) and The null assignment reflect (Section 3.3).

---

### Review · Reviewer_Bg2p · 2026-07-08

**Summary Of Contributions:**

this work tackles the issue stochastic dynamics in video world models.
in particular, authors provide a benchmarking protocol to study how faithful these models are.
they consider 6 sota image-to-video models, and 9 experimental scenes.
model's performance is decomposed into 2 sub-metrics: scorability - fraction of generation  producing a valid video, and calibration - the proximity of the outcomes distribution from the real/reference one.
a concise protocol is set in place to automatically conduct these experiments and report the performance.
the results suggest that the studied models fail to behave as a reliable stochastic simulator. in the case when the generation is realistic, the outcomes distribution across the videos does not match the real one - suffers from severe probability mass concentration and mode collapse.

strengths:
- the paper discusses an important issue, calibration in world models.
- it proposes a concise protocol to assess the generated videos calibration by assessing the outcomes distribution.
- it provides initial results over some specific simulations.

limitations:
- the calibration aspect focuses only on the outcome distribution and ignores how the outcome is generated. in short, only the outcome counts, but, how it is generated is not considered. one can achieve ideal outcome distribution in a unrealistic way.

- the studied scenes are limited only to outcomes with known reference distributions. this limits the application of the introduced 2-axes performance to real world video generation.

- the outcome extraction in the benchmark is done automatically via a model - gemini. while it has agreement of 93.2 over half of the generated videos, it still hold potential systematic errors that can falsify the conclusions of this work.

**Audience:**

Yes

**Audience Explanation:**

it falls within the topics of the journal.

**Broader Impact Concerns:**

no ethical concerns.

**Claims And Evidence:**

Yes

**Claims Explanation:**

the obtained results support the claim of poor calibration in world models. however, the simulated videos and outcome extraction still needs human validation as it is done by model in the current form.

**Requested Changes:**

minor: fig.1 should have a better elaborated caption.